# Approaching isotropic charge transport of n-type organic semiconductors with bulky substituents

Craig P. Yu [1], Naoya Kojima[2], Shohei Kumagai[1], Tadanori Kurosawa[1,2], Hiroyuki Ishii [3], Go Watanabe [4], Jun Takeya [1,2,5,6] & Toshihiro Okamoto [1,2,5,7,8 ✉]

Benzo[de]isoquinolino[1,8-gh]quinolinetetracarboxylic diimide (BQQDI) is an n-type organic semiconductor that has shown unique multi-fold intermolecular hydrogen-bonding interactions, leading to aggregated structures with excellent charge transports and electron mobility properties. However, the strong intermolecular anchoring of BQQDI presents challenges for fine-tuning the molecular assembly and improving the semiconducting properties. Herein, we report the design and synthesis of two BQQDI derivatives with phenyl- and cyclohexyl substituents (Ph–BQQDI and $Cy_6$–BQQDI), where the two organic semiconductors show distinct molecular assemblies and degrees of intermolecular orbital overlaps. In addition, the difference in their packing motifs leads to strikingly different band structures that give rise to contrasting charge-transport capabilities. More specifically, $Cy_6$–BQQDI bearing bulky substituents exhibits isotropic intermolecular orbital overlaps resulting in equal averaged transfer integrals in both π-π stacking directions, even when dynamic disorders are taken into account; whereas Ph–BQQDI exhibits anisotropic averaged transfer integrals in these directions. As a result, $Cy_6$–BQQDI shows excellent device performances in both single-crystalline and polycrystalline thin-film organic field-effect transistors up to 2.3 and 1.0 $cm^2 V^{-1} s^{-1}$, respectively.

[1] Material Innovation Research Center (MIRC) and Department of Advanced Materials Science, School of Frontier Sciences, The University of Tokyo, 5-1-5 Kashiwanoha, Kashiwa, Chiba 277-8561, Japan. [2] Department of Applied Chemistry, Faculty of Engineering, The University of Tokyo, 7-3-1 Hongo, Bunkyo-ku, Tokyo 113-0033, Japan. [3] Department of Applied Physics, Faculty of Pure and Applied Sciences, University of Tsukuba, 1-1-1 Tennodai, Tsukuba, Ibaraki 305-8573, Japan. [4] Department of Physics, School of Science, Kitasato University, 1-15-1 Kitasato, Minami-ku, Sagamihara, Kanagawa 252-0373, Japan. [5] National Institute of Advanced Industrial Science and Technology (AIST)-University of Tokyo Advanced Operando-Measurement Technology Open Innovation Laboratory (OPERANDO-OIL), AIST, 5-1-5 Kashiwanoha, Kashiwa, Chiba 277-8561, Japan. [6] International Center for Materials Nanoarchitectonics (MANA), National Institute for Materials Science (NIMS), 1-1 Namiki, Tsukuba 205-0044, Japan. [7] PRESTO, JST, 4-1-8 Honcho, Kawaguchi, Saitama 332-0012, Japan. [8] CREST, JST, 4-1-8 Honcho, Kawaguchi, Saitama 332-0012, Japan. ✉email: tokamoto@k.u-tokyo.ac.jp

Charge transport that gives rise to electrical properties of organic semiconductors (OSCs) is typically governed by intermolecular orbital overlaps, and controlling such intermolecular interactions to achieve effective charge-transport properties lies in the center of molecular design for high-performance OSCs[1,2]. In the past decades, intense investigations of high-performance OSCs in terms of molecular design and device engineering fueled the rapid development of applicable organic-based electronic devices such as organic field-effect transistors (OFETs)[3–5], which offer mechanical flexibility and low-cost processing compared with traditional inorganic-based devices. In particular, the hole-transporting p-type OSCs have shown promising OFET performances with charge-carrier mobilities ($\mu$) over $10 \, cm^2 \, V^{-1} \, s^{-1}$. Not only do these materials lead to applicable devices, but they also provide crucial information on charge transport and guidance for future molecular designs[6–13]. On the other hand, the electron-transporting n-type OSCs, which are an essential component for constructing organic-based logic circuits[14–16], are generally inferior to state-of-the-art p-type OSCs in terms of $\mu$. One of the challenges associated with the molecular design of n-type OSCs is that the excited molecules transporting injected charge carriers can be oxidized by ambient singlet oxygen and moisture, which leads to degraded electronic performances in air. Thus, the lowest unoccupied molecular orbital (LUMO) level of n-type OSCs should be below $-4.0 \, eV$ to avoid oxidation of charge carriers and ensure air-stable electron-transporting performances[17,18]. While the air-stability issue of n-type OSCs can be addressed by incorporations of electron-deficient moieties[19–21] and several studies have reported air-stable n-type OSCs with encouraging OFET performances[22–25], design strategies that focus on effective intermolecular orbital overlaps (quantified by transfer integral $t$ and effective mass $m^*$)[26,27] and molecular assemblies for achieving favorable charge-transport properties and high electron mobility ($\mu_e$) are still required.

Recently, our group reported an air-stable and high-performance benzo[de]isoquinolino[1,8-gh]quinolinetetracarboxylic diimide (BQQDI) $\pi$-electron core[28–30] ($\pi$-core) (Fig. 1a). The BQQDI is structurally analogous to the widely studied perylenetetracarboxylic diimide (PDI) system[31–34] (Fig. 1a), though the electronegative nitrogen atoms in the BQQDI framework result in a DFT-calculated deep-lying LUMO level of $-4.17 \, eV$ (at the B3LYP/6-31 G + (d) level[35]) for potential air-stable n-type charge transports. In contrast, the alkylated-PDI $\pi$-core possesses a shallower LUMO level of $-3.80 \, eV$. Upon functionalization of the BQQDI $\pi$-core with phenethyl (PhC$_2$–BQQDI) groups, multifold hydrogen-bonding interactions are formed between adjacent molecules in the transverse direction (Fig. 1b), and strong $\pi$–$\pi$ interactions are also observed in the $\pi$–$\pi$ stacking direction. The resulting brickwork-packing motifs show large $t$ values (Fig. 1c), which indicate two-dimensional (2D) charge-transport properties, whereas simple PDI (C$_8$–PDI, as an example) derivatives generally exhibit one-dimensional (1D) $\pi$–$\pi$ stacking motif[36,37] that leads to anisotropic charge-transport capabilities. PhC$_2$–BQQDI forms favorable phenyl-to-phenyl edge-to-face interactions between each molecular layer (Fig. 1b), in addition to the aforementioned intermolecular features, which significantly reinforce the intermolecular orbital overlaps as well as suppression of dynamic disorder. As a result, PhC$_2$–BQQDI exhibits an impressive $\mu_e$ of $3.0 \, cm^2 \, V^{-1} \, s^{-1}$ in solution-processed OFETs, and excellent robustness against thermal- and bias stress, which are necessary features for practical organic electronic applications. Despite the encouraging results of PhC$_2$–BQQDI as an n-type OSC, the robust core-to-core and interlayer intermolecular interactions also pose challenges to further fine-tune molecular assemblies and charge-transport properties of BQQDI derivatives. By examining the packing structure of

PhC$_2$–BQQDI, we notice that the hydrogen-bonding interactions cause some degree of $\pi$–$\pi$ stacking misalignment in both the long and short molecular axes (Fig. 1c), causing an unbalanced charge-transport capability reflected by its $t$ and $m^*$ values.

Herein, we report the investigation of two BQQDI derivatives with phenyl and cyclohexyl substituents (Ph–BQQDI and Cy$_6$–BQQDI, respectively) on their molecular assemblies and charge-transport capabilities. From a chemical perspective, we envisage that the installment of these sterically demanding (used with bulky interchangeably) substituents close to the BQQDI $\pi$-core compared with PhC$_2$–BQQDI may sufficiently weaken the hydrogen-bonding interactions in the transverse direction and reduce the misalignment in intermolecular orbital overlaps. Owing to the different geometric and electronic properties of Ph and Cy$_6$ substituents, Ph–BQQDI and Cy$_6$–BQQDI exhibit distinct intra- and interlayer molecular assemblies that lead to contrasting charge-transport capabilities as well as OSC performances.

## Results and discussion

**Synthesis**. The first target compound Ph–BQQDI was synthesized from the benzo[de]isoquinolino[1,8-gh]quinolinetetracarboxylic dianhydride (BQQ–TCDA) starting material according to the previously reported procedure[28] in 91% yield (Fig. 2a). However, formation of Cy$_6$–BQQDI could only reach 70% from BQQ–TCDA, along with 5% monofunctionalized intermediate and 25% remaining starting material, likely due to the low reactivity of amine with the bulky cyclohexyl moiety. Previously, the synthesis of 4-heptyl-substituted BQQDI (4-Hep–BQQDI) with bulky branched alkyl chains afforded only 10% yield by using BQQ–TCDA as the starting material. To circumvent this issue, we discover that the precursor of BQQ–TCDA, 3,9-dimethyl-4,10-bis(2,4,6-trichlorophenyl)benzo[de]isoquinolino[1,8-gh]quinoline-3,4,9,10-tetracarboxylate (BQQ–TC) can also act as a viable starting material for the synthesis of BQQDI derivatives. The electrophilic trichlorophenyl ester groups of BQQ–TC provide high reactivity[38] and tolerance to the somewhat bulky cyclohexyl amine. The Cy$_6$–BQQDI target compound was successfully furnished from BQQ–TC in 94% yield (Fig. 2b), and 4-Hep–BQQDI was also generated in 86% yield using the same procedure. Ph– and Cy$_6$–BQQDI exhibited high 5% weight-loss temperatures (Supplementary Fig. 4), as well as experimental LUMO energy levels below $-4.0 \, eV$ (Supplementary Fig. 5), which suggested thermally stable OSCs and air-stable electron transport in OFET operations.

**Molecular assemblies and charge transports**. Large plate-like single crystals of Ph–BQQDI and Cy$_6$–BQQDI were prepared using physical vapor transport and solution-grown methods, respectively (Supplementary Fig. 6 and Supplementary Data 1). Single crystals reported in this work were measured at room temperature (Supplementary Table 2). Ph–BQQDI crystallizes in the monoclinic P2$_1$/c space group with a 2D brickwork-packing motif. Each planar BQQ $\pi$-core forms multifold hydrogen-bonding interactions[39] with O···H and N···H close contacts on each side with its adjacent molecules in the transverse direction, along with misaligned $\pi$–$\pi$ stacking interactions (Fig. 3a). Within the brickwork assembly of Ph–BQQDI, distances of the $\pi$-stacks are found to be 3.36 Å and 3.37 Å (Fig. 3b), and the slight difference in distances is attributed to the misalignment between adjacent molecules in the transverse direction. The molecular assembly of Ph–BQQDI leads to a misalignment of LUMO in the $\pi$–$\pi$ stacking direction (Fig. 3a), where only a small degree of LUMO overlaps is observed between the top molecule and the molecule in the bottom layer. By calculating the $t$ values of Ph–BQQDI based on its crystal structure, it is evident that the

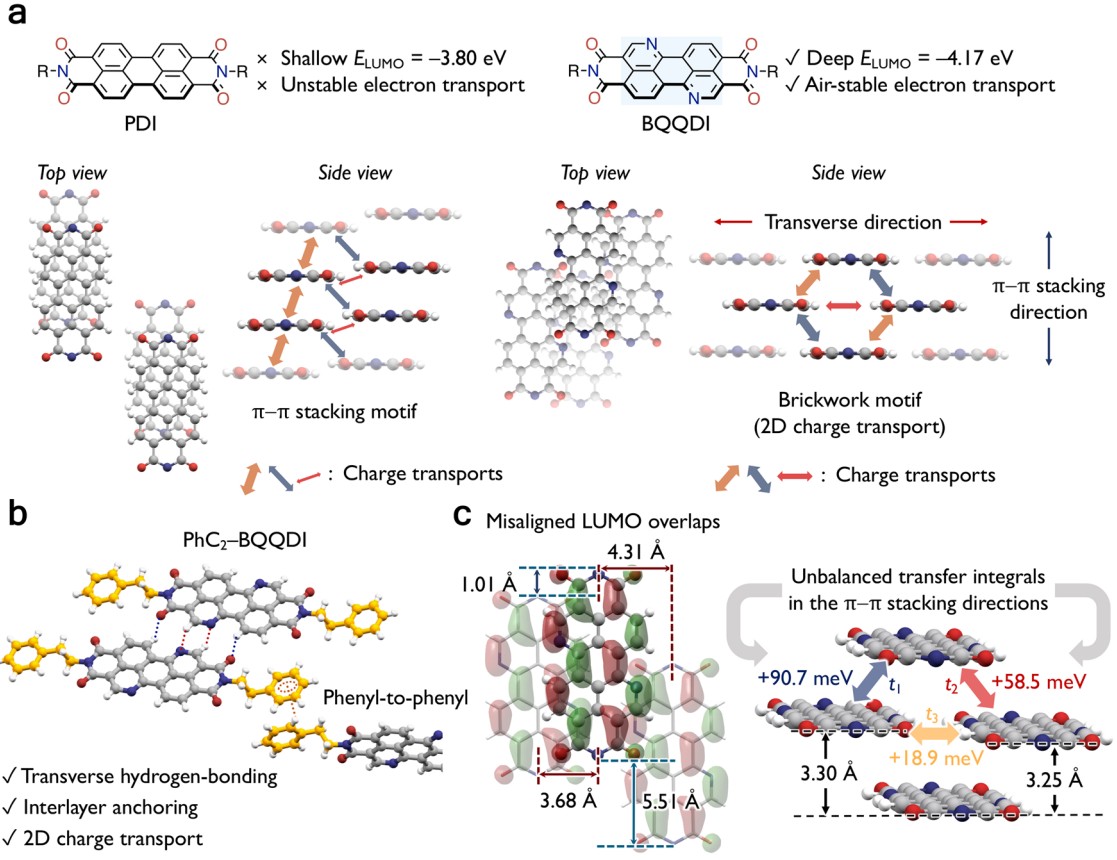

**Fig. 1 Molecular features of C₈–PDI and PhC₂–BQQDI. a** Structural, packing motif, and charge-transport comparisons between C₈–PDI and BQQDI (LUMO energy is calculated at the B3LYP/6-31 G + (d) level of theory, and arrow thickness represents the relative magnitude of transfer integrals). **b** Intermolecular interactions of PhC₂–BQQDI (orange and blue arrows indicate π–π stacking interactions, and the red arrow indicates transverse interactions). **c** Molecular misalignment, transfer integrals, and π–π stacking distances (between planes of atoms on the BQQ core, excluding hydrogens) of PhC₂–BQQDI.

**Fig. 2 Synthetic routes for BQQDI derivatives. a** Synthesis of Ph–BQQDI from BQQ–TCDA. **b** Synthesis of Cy₆–BQQDI from BQQ–TC.

misalignment in the assembly leads to different degrees of orbital overlaps with $t_1$ and $t_2$ equal to +78.4 and +49.1 meV, respectively (Fig. 3b). Strong transverse interactions between π-cores are quantified by $t_3$ values of +17.7 meV. By comparing the $t$ values of Ph–BQQDI with the high-performance PhC₂–BQQDI (Fig. 1c), the $t$ values of Ph–BQQDI are much smaller than those of PhC₂–BQQDI, especially in the π–π stacking direction.

Cy₆–BQQDI crystallizes in the monoclinic $C2/m$ space group with more symmetry than that of Ph–BQQDI (Supplementary Data 2). Different from Ph–BQQDI, the single-crystal structure of Cy₆–BQQDI exhibits a static disordering where nitrogen atoms can be found at different *bay* positions, where 50% occupancies were assumed. Although they can be randomly arranged in the actual structure, two types of periodic structures, namely, the A- and B

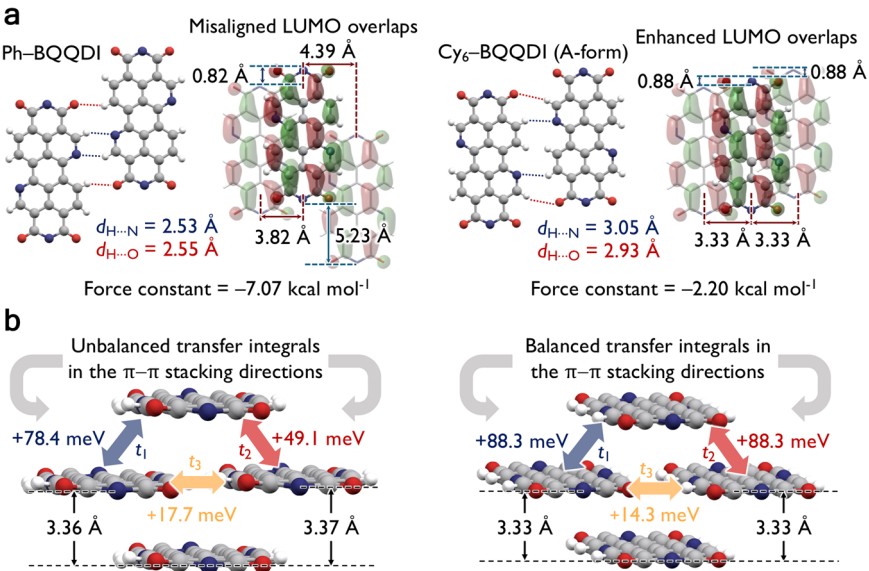

**Fig. 3 Molecular assemblies and charge-transport capabilities of Ph–BQQDI and Cy₆–BQQDI. a** Intermolecular distances and force constants between dimers along the transverse direction, molecular misalignment distances, and illustration of LUMO overlaps along the π–π stacking direction. **b** Illustration of the 2D brickwork molecular assembly, including π-π stacking distances (between planes of atoms on the BQQ core, excluding hydrogens) and the calculated transfer integrals.

forms (Fig. 3 and Supplementary Fig. 8a) were considered for the following computational analyses. We will first investigate the A form here. The steric bulk of cyclohexyl substituents likely prevents close contact between molecules in the transverse direction, which shows O···H and N···H distances of 2.93 Å and 3.05 Å, respectively, that are larger in distances than those between Ph and BQQDI in the transverse direction. Force-constant calculations of the transverse dimers at the M06-2X/6-31 ++G(d,p) level[40] further substantiate that Cy₆–BQQDI shows a much weaker interaction energy of –2.20 kcal mol⁻¹ than that of Ph–BQQDI (–7.07 kcal mol⁻¹) (Fig. 3a). However, transverse dimers of Cy₆–BQQDI show a much smaller displacement in the long molecular axis direction than Ph–BQQDI dimers, and the reduced molecular misalignment of Cy₆–BQQDI leads to a much more enhanced LUMO overlaps in the π–π stacking directions. The 2D brickwork motif of Cy₆–BQQDI shows a uniform π–π stacking distance of 3.33 Å, which corresponds to the same degree of intermolecular orbital overlap with $t_1 = t_2 = +88.3$ meV, which is larger than those of Ph–BQQDI. Even though the transverse dimer of Cy₆–BQQDI demonstrates much weaker interaction energy than that of Ph–BQQDI dimer, the transverse intermolecular orbital overlap of Cy₆–BQQDI that is quantified by $t_3$ (+14.3 meV) is only slightly lower than that of Ph–BQQDI (+ 17.7 meV) (Fig. 3b). The B form of Cy₆–BQQDI exhibits very similar $t$ values as the A form, with $t_1 = t_2 = +85.2$ meV and $t_3 = + 17.2$ meV (Supplementary Fig. 8a). The uniform charge-transport capability exhibited by Cy₆–BQQDI may indicate promising OSC performances[41].

**Dynamic disorders**. The different substituent effects of Ph–BQQDI and Cy₆–BQQDI prompted us to investigate their dynamic disorders in the single-crystal state, and MD simulations with constant number of molecules (N), temperature (T), and pressure (P) (isothermal–isobaric NTP ensemble) are performed based on their single-crystal structural data at room temperature (Fig. 4, Supplementary Table 3, and Supplementary Data 3–5). Ph–BQQDI shows small B-factors, which is the thermal factor for each atom (see "Method" section for the mathematical definition), where B-factors are observed on the substituents as well as the π-cores. However, the atoms on the *ortho* and *bay* positions[42]

(Fig. 4a) of Ph–BQQDI molecules show slightly larger B-factors than the rest of the π-core, which may affect the charge transport in the π–π stacking directions. Since A- and B forms of the crystal structure are assumed for Cy₆–BQQDI, we examined their MD simulations separately. Interestingly, A- and B forms of Cy₆–BQQDI demonstrate drastically different degrees of molecular fluctuations, with the A form showing small B-factors similar to those of Ph–BQQDI (Fig. 4a), while the B form exhibits much larger B-factors that indicate larger disorders(Supplementary Fig. 9).

We picked up 100–200 pairs of adjacent dimers in the π–π stacking directions from the MD-simulated Ph– and Cy₆–BQQDI. Variant $t_1$ and $t_2$ distributions and standard deviations (σ) are calculated to reveal the effect of dynamic disorders on charge-transport capabilities. Ph–BQQDI has an averaged $t_1 = +59.7$ meV and $t_2 = +34.2$ meV, with corresponding σ of 24.2 and 11.8 meV, respectively (Fig. 4b). To our surprise, despite having completely different B-factors, A- and B forms of Cy₆–BQQDI demonstrate very similar variant $t$ values in the π–π stacking directions. The A form is showing averaged $t_1 = +66.7$ meV and $t_2 = +69.1$ meV, with σ of 16.6 and 16.9 meV (Fig. 4b). The B form shows averaged $t_1 = +68.1$ meV and $t_2 = +69.5$ meV that are similar to those of the A form, despite the former's large B-factors. The σ of averaged $t_1$ and $t_2$ of the B form is calculated to be 21.6 and 24.1 meV (Supplementary Fig. 9). It has been reported that the ratio of σ and averaged $t$ values (σ/$t_{Avg.}$) quantify the dynamic disorder[43]. Ph–BQQDI exhibits σ/$t_{Avg}$ of 0.41 and 0.35 in $t_1$ and $t_2$ directions, respectively. Cy₆–BQQDI demonstrates smaller σ/$t_{Avg}$ of 0.25 and 0.24 for the A form, and 0.32 and 0.35 for the B form, in $t_1$ and $t_2$ directions, respectively. The current calculations suggest that the charge-transport capability of Ph–BQQDI is strongly affected by the dynamic disorder compared with Cy₆–BQQDI. Troisi *et al.* reported that large isotropic $t$ values in the 2D herringbone assembly can be insensitive toward the dynamic disorder[41]. Our results here may suggest that the isotropic $t$ values of Cy₆–BQQDI in the 2D brickwork assembly may also provide resilience to the dynamic disorder.

**Single-crystalline transistor performances**. To evaluate the μₑ of Ph– and Cy₆–BQQDI, we fabricated bottom-gate/top-contact

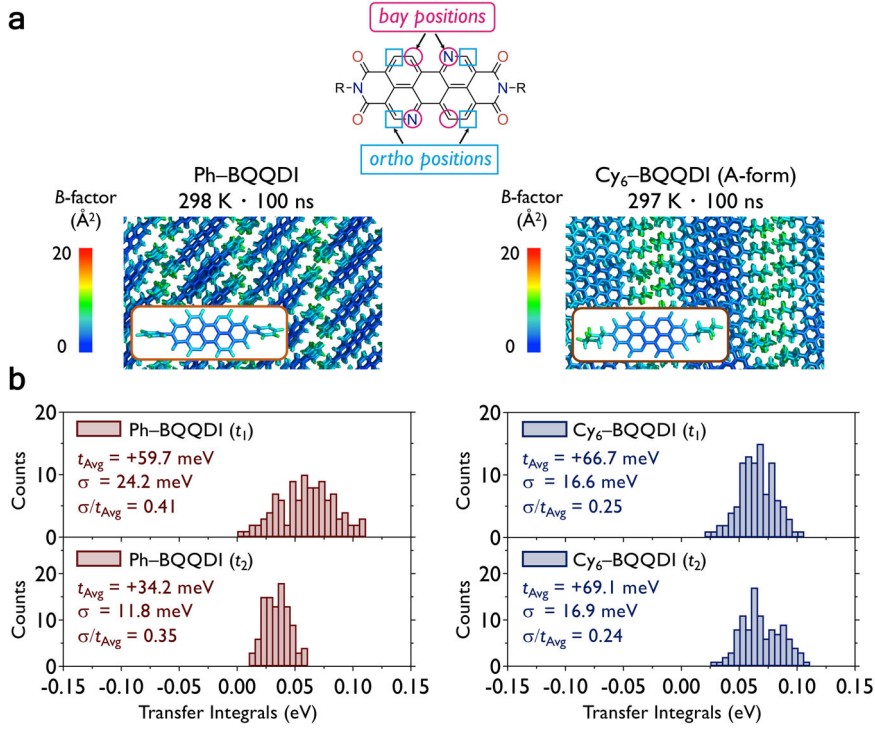

**Fig. 4 Molecular dynamics simulations of Ph–BQQDI and Cy$_6$–BQQDI. a** *Ortho/bay* positions of BQQDI and color-coded *B*-factor (Å$^2$) distributions obtained from the trajectories during the last 10 ns of a 100-ns MD simulation in the NTP ensemble (the magnitude of *B*-factors is represented by the color-coded scale bar ranging from blue (small value) to red (large value)). **b** Variant $t_1$ and $t_2$ distributions and standard deviations (σ) calculated from 100 pairs of adjacent dimers, revealing the magnitude of the dynamic disorders.

OFETs with gold electrodes using their single-crystalline thin films as the active OSC layer. Owing to the poor solubility of Ph–BQQDI, crystals grown by physical vapor transport (193-nm thick) were directly laminated on a silicon substrate coated with a parylene-insulating polymer, which has been used for laminated single-crystal OFETs[44]. The OSC single-crystalline thin films (7.8-nm thick) (Supplementary Fig. 13) of the more soluble Cy$_6$–BQQDI were prepared by the edge-casting method[45] on the AL-X601-coated silicon substrate, which is a common insulating layer for solution-processed BQQDI materials[28]. The maximum $\mu_e$ of Ph–BQQDI was measured to be 1.0 cm$^2$ V$^{-1}$ s$^{-1}$ (Fig. 5a). The highest $\mu_e$ of 2.3 cm$^2$ V$^{-1}$ s$^{-1}$ was achieved by Cy$_6$–BQQDI and an average $\mu_e$ of 1.8 ± 0.21 cm$^2$ V$^{-1}$ s$^{-1}$ was measured over 12 OFETs (Supplementary Fig. 10), and the devices showed excellent air stability over one month (Supplementary Fig. 11). The large threshold voltage and the nonideal transfer curve exhibited by Cy$_6$–BQQDI is possibly due to the contact resistance attributed to the disrupted molecular assembly at the electrode–OSC interface[46], which leads to a low reliability factor[47] ($r_{sat}$) of 0.29 (Supplementary Fig. 12), and an effective $\mu$ of 0.67 cm$^2$ V$^{-1}$ s$^{-1}$ (effective $\mu = r_{sat} \times \mu_{claimed}$, where $\mu_{claimed}$ is the primarily reported $\mu_e$). X-ray diffractions of Ph– and Cy$_6$–BQQDI thin films reveal that their OFET channel directions correspond to the *b*-crystallographic axis and the [110] direction, respectively (Supplementary Fig. 7). The molecular stacks of Cy$_6$–BQQDI are roughly orthogonal to the OFET substrate with the π–π stacking parallel to the electron transport direction. On the other hand, the molecular assembly of Ph–BQQDI creates more of an offset between the electron transport and the π–π stacking direction, which possibly leads to a less efficient electron transport similar to PhC$_2$–BQQDI.

**Polycrystalline transistor performances**. Polycrystalline thin-film (40-nm thick) OFETs of Ph– and Cy$_6$–BQQDI were also

fabricated via vacuum deposition using decyltrimethoxysilane (DTS) as the self-assembled monolayer. The deposited thin film of Ph–BQQDI does not assume its single-crystal structure, as the polycrystalline *d*-spacing of 19.5 Å at $2\theta = 4.52°$ differs from its single-crystalline *d*-spacing of 15.5 Å. A hypothesized tilting angle between the long axis of Ph–BQQDI and the substrate is 24.5° based on the longest intramolecular H···H distance (21.47 Å based on the single-crystal structure) (Supplementary Fig. 16 and 18), which possibly originates from the interactions between the substrate and OSC molecules. The polycrystalline thin film of Cy$_6$–BQQDI on the other hand, shows consistent molecular assembly with its single-crystal structure. Though the diffraction peak at $2\theta = 17.04°$ corresponds to the (11$\bar{1}$) plane of the single-crystal structure, which indicates a thin-film orientational disordering[48] with both edge-on and face-on-like stackings (Supplementary Fig. 16 and 19). We evaluated the polycrystalline thin-film OFETs, and the highest $\mu_e$ of 0.16 cm$^2$ V$^{-1}$ s$^{-1}$ was obtained for Ph–BQQDI (Supplementary Fig. 20a), which is one order lower than its single-crystalline device. Although the critical reason has not been clarified, we hypothesize that the inconsistent polycrystalline thin-film assembly with its single-crystal structure possibly leads to a poorer electron-transport capability in the former. In addition, the surface morphology of Ph–BQQDI with less significant terracing structure than that of the Cy$_6$–BQQDI thin film, despite comparable grain sizes (>500 nm), implies lower crystallinity of Ph–BQQDI thin films (Supplementary Fig. 14). Cy$_6$–BQQDI-based polycrystalline OFETs afforded the highest $\mu_e$ of 0.50 cm$^2$ V$^{-1}$ s$^{-1}$ on DTS (Supplementary Fig. 21), and this promising result motivated us to explore other device conditions. When the self-assembled monolayer was changed from DTS to hexamethyldisilazane (HMDS), the ratio of face-on/edge-on assemblies was decreased (Supplementary Fig. 17a), and the highest $\mu_e$ of 40-nm-thick polycrystalline devices of Cy$_6$–BQQDI was further improved to 0.66 cm$^2$ V$^{-1}$ s$^{-1}$ (Supplementary

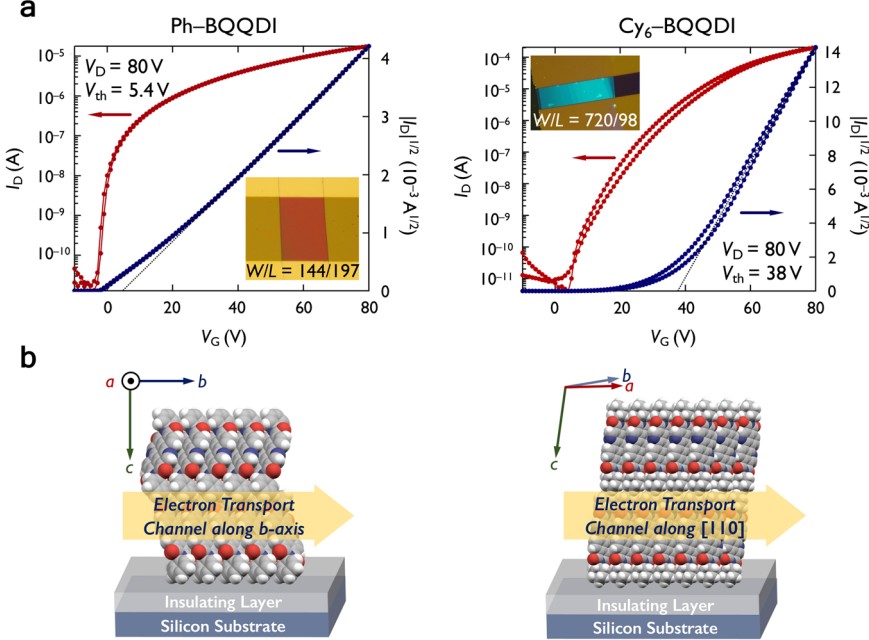

**Fig. 5 OFET performances and thin-film assemblies of Ph–BQQDI and Cy₆–BQQDI. a** Optical microscopic images and transfer characteristics of single-crystalline transistors (black dashed lines represent the fit to $|I_D|^{1/2}$, from which the $\mu_e$ are estimated), where $V_D$ and $V_{th}$ are drain and threshold voltages. **b** Molecular assemblies in device states and the corresponding channel directions.

Fig. 22). We found that by reducing the Cy₆–BQQDI OSC-layer thickness from 40 nm to 20 nm on HMDS, an excellent highest $\mu_e$ of 1.0 cm² V⁻¹ s⁻¹ was achieved (Supplementary Fig. 23), which implied that homogeneous films with less terracing features (Supplementary Fig. 15) afforded by reducing the film thickness were preferred for charge transport. The device performances of polycrystalline Cy₆–BQQDI on DTS and HMDS in air are also consistent over more than one month (Supplementary Fig. 24). The $\mu_e$ of polycrystalline Cy₆–BQQDI is one of the highest among current BQQDI derivatives (the highest polycrystalline $\mu_e$ of PhC₂–BQQDI is 0.65 cm² V⁻¹ s⁻¹)[29], though, we speculate that its overall polycrystalline device performance might be hampered by the orientational disordering of its thin-film molecular assembly, and further optimization of the deposition conditions is currently undergoing. On the other hand, the polycrystalline Ph–BQQDI OFETs on HMDS resulted in lower $\mu_e$ by an order of magnitude (0.024 cm² V⁻¹ s⁻¹) (Supplementary Fig. 20b) due to lowered crystallinity with the same aggregated structure (Supplementary Fig. 16 and 17b). Both single- and polycrystalline OFETs based on Cy₆–BQQDI show significantly higher $\mu_e$ than those based on Ph–BQQDI, and the difference in their device performances is in agreement with their calculated $t$ values, but more in-depth analysis of their charge-transport capabilities is required.

**Estimations of effective mass.** We then further investigated the directionality of charge-transport capabilities of Ph– and Cy₆–BQQDI using the tight-binding approximation[49] to rationalize the difference in their OFET performances. Ph–BQQDI exhibits elliptical-shaped 2D LUMO bands, and from the band dispersion (Fig. 6a and Supplementary Fig. 25), we calculated the $m^*$ values with respect to the crystallographic axes. The smallest $m^*$ value of Ph–BQQDI is 1.6 $m_0$ ($m_0$: the rest mass of an electron), which is found at 45° from the $a$-axis ([110] direction) (Fig. 6b). However, the OFET channel along the $b$-axis direction of Ph–BQQDI corresponds to a larger $m^*$ of 2.4 $m_0$. On the other hand, Cy₆–BQQDI shows a more circular 2D LUMO band and the resulting $m^*$ values are seemingly uniform along all crystallographic directions (Fig. 6d). The $m^*$ of Cy₆–BQQDI along the

OFET channel direction is estimated to be 1.9 $m_0$, which is only slightly larger than its smallest $m^*$ of 1.8 $m_0$ in the $a$-axis direction (Fig. 6e, $m^*$ of the B-form is shown in Supplementary Fig. 8b). The directionality of charge transport can be better visualized from the angle-resolved inversed $m^*$ plots, where Ph–BQQDI shows a peanut-shaped curve, with its best charge transport at 45° relative to the $a$-axis ([110] direction), and poorer charge-transport capability is found along the $b$-axis, which is the channel direction (Fig. 6c). The inversed $m^*$ plot of Cy₆–BQQDI shows a more uniform charge transport, where a favorable charge-transport capability can be found along the channel direction of its OFETs (Fig. 6f). We have further confirmed the isotropic-like charge-transport capability of Cy₆–BQQDI by evaluating the $\mu_e$ of its single-crystalline device at 0°, 45°, –45°, and 90° relative to the crystal-growth direction [110], and the $\mu_e$ are found in the range of 1.5–2.0 cm² V⁻¹ s⁻¹ (Supplementary Fig. 26 and 27). The high polycrystalline-device performance of Cy₆–BQQDI, despite having the orientationally disordered thin-film assembly, could be attributed to its isotropic-like charge-transport capability. It could also explain its superior polycrystalline-device performances to those of our previously reported phenylalkyl-substituted BQQDI derivatives fabricated under similar conditions that exhibit more pronounced anisotropic charge-transport behaviors[29]. The current results suggest the importance of molecular design not only in the bulk-crystal state, but also in the thin-film state along the channel direction for achieving high device performances.

**Conclusions**
In summary, the current work reports distinct effects of two sterically demanding substituents on the molecular assemblies in both bulk crystal and thin-film states. Their molecular assemblies lead to different charge-transport capabilities, where Cy₆–BQQDI exhibits uniform transfer integrals and effective mass compared with Ph–BQQDI. Intriguingly, Cy₆–BQQDI with isotropic charge transport exhibits resilience to the dynamic disorders, which is superior to the high-performance PhC₂–BQQDI in this regard. From the tight-binding approximations, the smallest effective

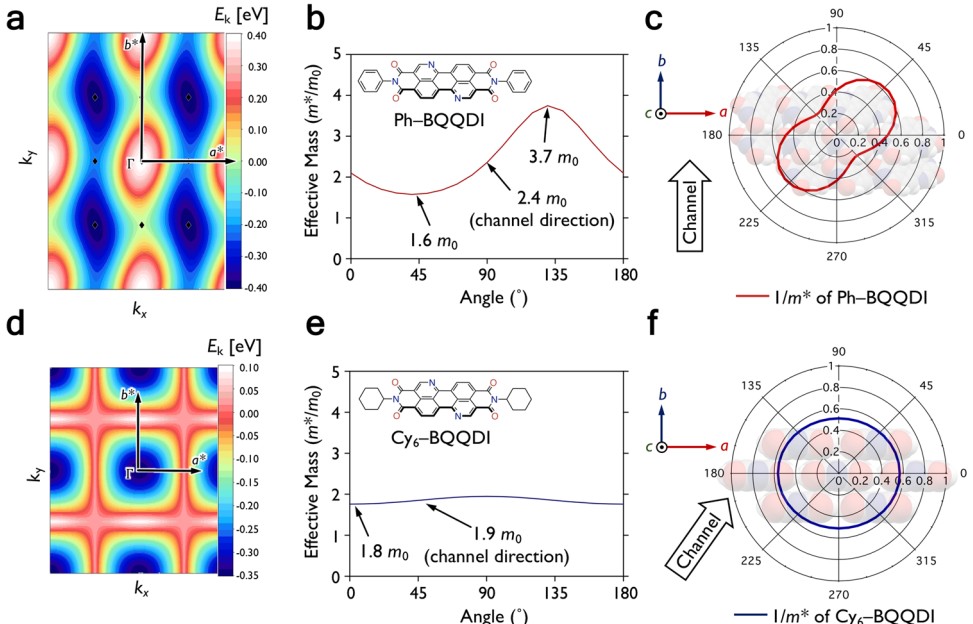

**Fig. 6 Charge-transport capabilities calculated by the tight-binding approximation. a** and **d** Contour plots of 2D HOMO bands, where the origin of the energy axis is set to the LUMO level. **b** and **e** Angle-resolved effective mass plots. **c** and **f** Angle-resolved inversed effective mass plots and molecular assemblies in the *ab*-plane of Ph–BQQDI and Cy₆–BQQDI, respectively.

mass of Ph–BQQDI is smaller than that of Cy₆–BQQDI. However, the effective mass of Cy₆–BQQDI along the OFET channel direction is smaller than that of Ph–BQQDI. Thus, OFETs of Cy₆–BQQDI exhibit an excellent $\mu_e$ of 2.3 cm² V⁻¹ s⁻¹ in single-crystalline thin films, and up to 1.0 cm² V⁻¹ s⁻¹ in polycrystalline devices. The results herein demonstrate an effective molecular design for molecular assembly, charge transport, and suppressing dynamic disorders in the bulk single-crystal state, as well as for controlling the molecular assembly in the thin-film device state for achieving high device performances via substituent engineering. Future work based on the current encouraging results of Cy₆–BQQDI may involve incorporations of alkyl substituents on the cyclohexyl group to improve its solubility for large-area device fabrications.

## Methods
**Materials and general characterizations**. All amine reagents used in this study were purchased from Tokyo Chemical Industry Co., Ltd and propionic acid was purchased from FUJIFILM Wako Pure Chemical Industries, Ltd., without further purifications. *o*-Dichlorobenzene (*o*-DCB) was purchased from KANTO chemical Co., Ltd., and purified by a solvent-purification system. Starting materials BQQ–TCDA and BQQ–TC were synthesized and purified in our laboratory prior to this study. All reactions were carried out under an atmosphere of argon. ¹H NMR spectra were recorded on JEOL ECS400 spectrometer (400 MHz). Chemical shifts were reported in parts per million (ppm, δ scale) from residual protons in the deuterated solvent for ¹H NMR (5.93 ppm for 1,1,2,2-tetrachloroethane-*d₂* (TCE-*d₂*), 3.36 ppm/4.37 ppm for 1,1,1,3,3,3-hexafluoro-2-propanol-*d₂* (HFIP-*d₂*), and 7.26 ppm for chloroform-*d* (CDCl₃)). The data were presented in the following format: chemical shift, multiplicity (s = singlet, d = doublet, t = triplet, quint = quintet, m = multiplet), coupling constant in hertz (Hz), and signal-area integration in natural numbers. Time-of-flight high-resolution mass (TOF-MS) spectrometry measurements were measured on a BRUKER compact-TKP2 mass spectrometer with the atmospheric-pressure chemical ionization (APCI) method. Elemental analysis measurements were carried out on a JScience Lab JM10 CHN analyzer at the Comprehensive Analysis Center, the Institute of Scientific and Industrial Research, Osaka University. Differential pulse voltammetry (DPV) measurements were performed on an ALS622D Electrochemical Analyzer using glassy carbon as the working electrode, platinum as the counter electrode, and 0.01 M AgNO₃ + 0.1 M tetrabutylammonium hexafluorophosphate (TBAPF₆) in benzonitrile as the reference electrode. Compounds were dissolved in benzonitrile at 100 °C (<0.2 mM, saturated solution) and the DPV measurements were performed at the same temperature at a scan rate of 100 mV s⁻¹, with 0.1 M TBAPF₆ as the electrolyte and ferrocene as an internal standard.

**Synthetic procedure for Ph–BQQDI**. A flame-dried Schlenk tube was charged with BQQ–TCDA (117 mg, 0.297 mmol, 1.0 equiv.), aniline (275 mg, 2.96 mmol, 10.0 equiv.), propionic acid (2.00 mL, 297 mmol, 100 equiv.), and *o*-DCB (10.0 mL), and the mixture was stirred at 150 °C for 20 h under an atmosphere of argon. The resulting mixture was cooled to room temperature and precipitated in MeOH. The product was collected via vacuum filtration as a red solid (150 mg, 91% crude yield). **¹H NMR** (400 MHz, TCE-*d₂*): δ 9.70 (s, 2H), 9.36 (d, *J* = 8.0 Hz, 2H), 8.90 (d, *J* = 8.0 Hz, 2H), 7.62–7.33 (m, 10H). ¹³C NMR spectrum could not be obtained due to insufficient solubility. **HRMS** (APCI⁺-TOF): Calcd for C₃₄H₁₆N₄O₄ [M + H] 545.1250, found 545.1271. **Elemental analysis**. Calcd for C₃₄H₁₆N₄O₄: C, 75.00; H, 2.96; N, 10.29. Found: C, 74.92; H, 2.94; N, 10.12.

**Synthetic procedure for Cy₆–BQQDI**. A flame-dried Schlenk tube was charged with BQQ–TC (150 mg, 0.184 mmol, 1.0 equiv.), cyclohexylamine (54.6 mg, 0.551 mmol, 3.0 equiv.), and *o*-DCB (6.1 mL) and the reaction mixture was stirred at 150 °C for 1 h under an atmosphere of argon. After that, propionic acid (1.38 mL, 18.4 mmol, 100 equiv.) was added and the mixture was stirred at 150 °C for 3 h. The resulting mixture was cooled to room temperature and poured into a stirring solution of MeOH. The precipitates were collected via vacuum filtration to give the product as a red–purple solid (96.6 mg, 94% crude yield). **¹H NMR** (400 MHz, CDCl₃/HFIP-*d₂*): δ 9.58 (s, 2H), 9.30 (d, *J* = 7.6 Hz, 2H), 8.85 (d, *J* = 8.0 Hz, 2H), 5.04–4.96 (m, 2H), 1.97–1.764 (m, 16H), 0.91–0.86 (m, 4H). ¹³C NMR spectrum could not be obtained due to insufficient solubility. **HRMS** (APCI⁺-TOF): Calcd for C₃₄H₂₈N₄O₄ [M + H] 557.2189, found 557.2203 **Elemental analysis**. Calcd for C₃₄H₂₈N₄O₄: C, 73.37; H, 5.07; N, 10.07, found: C, 73.24; H, 5.13; N, 9.99.

**Synthetic procedure for 4-Hep-BQQDI**. A flame-dried Schlenk tube was charged with BQQ–TC (100 mg, 0.122 mmol, 1.0 equiv.), 4-heptylamine (42.3 mg, 0.367 mmol, 3.0 equiv.), propionic acid (0.912 mL, 12.2 mmol, 100 equiv.), and *o*-DCB (4.00 mL), the reaction mixture was stirred at 150 °C for 1 h under an atmosphere of argon. The resulting mixture was cooled to room temperature and poured into a stirring solution of MeOH. The precipitates were collected via vacuum filtration to give the product as a deep-red solid (61.7 mg, 86% crude yield). **¹H NMR** (400 MHz, CDCl₃/HFIP-*d₂*): δ 9.61 (s, 2H), 9.25 (d, *J* = 7.6 Hz, 2H), 8.81 (d, *J* = 8.0 Hz, 2H), 5.22–5.15 (m, 2H), 2.25–2.19 (m, 4H), 1.89–1.84 (m, 4H), 1.37–1.34 (m, 8H), 0.95–0.92 (m, 12H). **HRMS** (APCI⁺-TOF): Calcd for C₃₆H₃₆N₄O₄ [M + H] 589.2815, found 589.2843. **Elemental analysis**. Calcd for C₃₆H₃₆N₄O₄: C, 73.45; H, 6.16; N, 9.52, found: C, 73.23; H, 6.18; N, 9.39.

**Theoretical calculations**. Estimations of transfer integral and effective mass were conducted using the GAMESS package[50]. The Kohn–Sham eigenstates of all compounds in this work were calculated at the PBEPBE/6-31 G(d) level of theory. Transfer integrals between LUMOs of neighboring molecules in the crystal structures were estimated by the dimer method[1]. To further understand the charge-transport capabilities in the single-crystal state, their LUMO band structures $E(k)$

were calculated by the tight-binding approximation using transfer integrals. Intermolecular-interaction energy between two adjacent molecules was obtained at the M06-2X/6-31 + +G(d,p) level of DFT with counterpoise correction for the basis-set superposition error[40]. The calculations were performed using the Gaussian 09 program package[51].

**Thermal properties**. Thermogravimetric−differential thermal analysis (TG − DTA) was performed on a Rigaku Thermo Plus EVO II TG 8121 at a heating rate of $1 \, \mathrm{K \, min^{-1}}$ under a nitrogen flow of $100 \, \mathrm{mL \, min^{-1}}$.

**Solubility measurements**. To a weighed sample of around 1 mg was added 200 µL of 1-chloronaphthalene, repeatedly. The resulting suspension was shaken and heated at 150 °C, until complete dissolution. The total amount of solvent (mL) was converted into solubility in wt%.

**Molecular-dynamics simulations**. MD simulations of single-crystal structures in this study were carried out by using the MD program GROMACS 2016.3. The number of molecules, temperature, and the size and the shape of the initial MD cell of Ph-BQQDI and Cy$_6$-BQQDI are listed in the Supporting Information. Since the intra- and interatomic interactions should be treated explicitly for analyzing the atomistic dynamics, an all-atom model was employed in accordance with generalized Amber force-field parameters[52]. The partial atomic charges of the simulated molecules were calculated using the restrained electrostatic potential (RESP)[53] methodology, based on DFT calculations with the 6-31 G(d) basis set using the GAUSSIAN 09 revision E01 program[51].

For each system, the preequilibration run was initially performed at the given temperature for 5 ns after the steepest-descent energy minimization. All systems were subjected to preequilibration runs in the NTV (constant number of substances (N), constant temperature (T), and constant volume (V)) ensemble before their equilibration runs. During the preequilibration runs for the NTV ensemble, the Berendsen thermostat[54] was used to maintain the temperature of the system with relaxation time of 0.2 ps and the volume of the MD cell was kept constant. Subsequently, the NTP ensemble of the equilibration run was performed using the Nosé–Hoover thermostat[55–57] and Parrinello–Rahman barostat[58] with relaxation times of 1.0 and 5.0 ps, respectively. For all MD simulations in the NTP ensemble, the pressure of the system was kept at 1.0 bar. The smooth particle-mesh Ewald (PME)[59] method was employed to treat the long-range electrostatic interactions and the real-space cutoff and the grid spacing are 1.2 and 0.30 nm, respectively. The time step was set to 1 fs.

To compare temperature dependence of thermal atomic fluctuations between different molecules, we calculated the $B$-factors related to the thermal stability as expressed below:

$$B = \frac{8}{3} \pi^2 \Delta_i^2$$

where $\Delta_i$ is the root-mean-square fluctuations (RMSF) of atom $i$. The RMSF values can be estimated by using the following equation:

$$\Delta_i = \sqrt{\frac{1}{T} \sum_{j=1}^{T} |\boldsymbol{r}_i(t_j) - \bar{\boldsymbol{r}}_i|^2}$$

where $T$ is the number of steps, $\boldsymbol{r}_i(t_j)$ is the position coordinate of atom $i$, and $\bar{\boldsymbol{r}}_i$ is the average of $\boldsymbol{r}_i(t_j)$ during $T$. The RMSF values were analyzed from MD trajectories during the last 10 ns in the equilibrium.

More than one hundred pairs of dimers from the MD-simulated molecules are picked up to calculate their $t$ values in π–π stacking directions in response to dynamic disorders . Variant $t_1$ and $t_2$ values as well as their σ are calculated to show the effect of the dynamic disorders on charge-transport capabilities.

**X-ray crystallography**. Ph−BQQDI single crystals were obtained by means of physical vapor transport, and Cy$_6$-BQQDI crystals were grown in the mixture of nitrobenzene and 1-chloronaphthalene via the slow-cooling method. Single-crystal X-ray diffraction data were collected on a Rigaku R-AXIS RAPID II imaging-plate diffractometer with CuKα radiation ($\lambda = 1.54187$ Å) at room temperature. The structures were solved by direct methods [SHELXT (2015)] and refined by full-matrix least-squares procedures on F2 for all reflections [SHELXL (Ver. 2014/7)]. While positions of all hydrogen atoms were calculated geometrically, and refined by applying riding model, all other atoms were refined anisotropically. Polycrystalline thin-film X-ray diffractions were collected by $2\theta/\omega$ scan on a Rigaku SmartLab diffractometer with a CuKα source ($\lambda = 1.54056$ Å).

**Single-crystalline OFET fabrications and evaluationsh**. A highly n$^{++}$-doped silicon wafer was used as the substrate, with which the surface was treated by a fluorinated insulating polymer, AL-X601 for Cy$_6$-BQQDI. The highly n$^{++}$-doped silicon wafer with thermally grown SiO$_2$ layer (200 nm) was ultrasonicated in acetone and isopropanol, and then dried on a hotplate in air. Following UV−O$_3$ treatment, AL-X601 diluted with propylene glycol monomethyl ether acetate (PGMEA) was spin-coated onto the wafer and baked at 150 °C for 5 min in air, followed by curing at 180 °C for 10 min. Preparations of single-crystalline thin films were carried out by the solution-processed edge-casting method. Thin-film crystals of Cy$_6$-BQQDI were grown from 0.015 wt% 1-chloronaphthalene solution at 140 °C. After the completion of crystallization, thin films were thoroughly dried in a vacuum oven at 100 °C for 10 hours. Thickness of the thin films was determined by atomic force microscopy. Then, 40-nm-thick gold layers were vacuum-deposited through a metal shadow mask, acting as source and drain electrodes. Objective-channel regions were edged by the conventional Nd:YAG laser-etching technique. Before measurements, thermal annealing at 100 °C for 10 hours prior to electrical evaluations. The gate capacitance per unit area ($C_i$) for the AL-X601-containing gate dielectrics was measured to be 12.5 nF cm$^{-2}$ by a Keithley 4200-SCS.

Single crystals of Ph−BQQDI were prepared by the physical vapor-transport technique with a two-zone furnace under an Ar flow at 80 cm$^3$ min$^{-1}$. High and low temperatures were set to 460 °C and 345 °C, respectively. Red-platelet crystals were manually laminated onto an n$^+$-Si/SiO$_2$ (200-nm) substrate encapsulated by a 200-nm-thick parylene layer, where n$^+$-Si and SiO$_2$/parylene acted as a gate electrode and a gate insulator, respectively. In all, 100-nm-thick Au layers, which served as source and drain electrodes, were vacuum-deposited onto the laminated single crystal through a metal shadow mask. The $C_i$ for the SiO$_2$/parylene was measured to be 8.83 nF cm$^{-2}$ by a Keithley 4200-SCS.

Electrical evaluations of OFETs were conducted on a Keithley 4200-SCS semiconductor parameter analyzer in air. Electron mobility and threshold voltage were extracted from the transfer characteristics by using the conventional equation for the saturation regime:

$$\sqrt{|I_\mathrm{D}|} = \sqrt{\frac{W \mu_\mathrm{sat} C_\mathrm{i}}{2L}} (V_\mathrm{G} - V_\mathrm{th}),$$

where $I_\mathrm{D}$ is the drain current, $W$ the channel width, $\mu_\mathrm{sat}$ the saturated electron mobility (reported as $\mu_\mathrm{e}$ in the main text), $C_\mathrm{i}$ the gate capacitance per unit area, $L$ the channel length, $V_\mathrm{G}$ the gate voltage, and $V_\mathrm{th}$ the threshold voltage.

**Vacuum-deposited polycrystalline thin-film fabrications and evaluations**. Vacuum-deposited 20- and 40-nm-thick polycrystalline thin films were used to produce top-contact, bottom-gate OFETs. For device fabrication, a highly n$^{++}$-doped silicon wafer with a thermally grown SiO$_2$ layer (200 nm) was used as a substrate, where SiO$_2$ surface was modified with either DTS or HMDS. Before surface modification, the silicon wafer was washed by ultrasonication in acetone and isopropanol. After drying on a hotplate in air, the wafer was treated with UV−O$_3$. For DTS treatment, the wafer was exposed DTS vapor at 130 °C for 3 h, whereas HMDS was spin-coated, followed by annealing on a hotplate at 110 °C for 5 min, for HMDS treatment. The DTS-modified wafer was washed in toluene, acetone, and isopropanol prior to use, whereas the HMDS-modified wafer was used immediately. Then, OSCs were vacuum-deposited at a rate of 0.5 Å s$^{-1}$ to form 40-nm-thick polycrystalline films, during which the substrates were kept at 180 °C. A gold coating was subsequently vacuum-evaporated through a shadow mask to obtain 60-nm-thick source and drain electrodes. Channel lengths ($L$) and widths ($W$) were 100 and 2000 µm, respectively, after patterning by the laser etching. Before evaluation, OFETs were thermally annealed at 60 °C for 10 h in a vacuum oven. Electrical evaluations of OFETs were conducted on a Keithley 4200-SCS semiconductor parameter analyzer in air. Electron mobility and threshold voltage were extracted from the transfer characteristics by using the conventional equation for the saturation regime.

**Polycrystalline thin-film morphology**. Atomic force microscope images were obtained using a Shimadzu SPM-9700HT instrument in dynamic mode.

## Data availability

The data reported in this study are available from the corresponding author (Toshihiro Okamoto; tokamoto@k.u-tokyo.ac.jp) upon reasonable requests. Crystallographic data have been deposited in the Cambridge Crystallographic Data Centre (CCDC) as a supplementary publication under accession nos. CCDC-1997507 (Cy$_6$−BQQDI, 297 K), CCDC-1997508 (Ph−BQQDI, 298 K). These data can be obtained free of charge at www.ccdc.cam.ac.uk/data_request/cif.

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

## Acknowledgements

The authors thank AGC Inc. for supplying AL-X601. The computation reported in this paper was performed at the Research Center for Computational Science, Okazaki, Japan.

This work was supported by the JST-PRESTO and JST-CREST programs "Scientific Innovation for Energy Harvesting Technology" (numbers JPMJPR17R2, JPMJCR21Q1) and by KAKENHI. C.P.Y thanks the Grant-in-Aid for JSPS Fellows (number JP20J12608), T.O., H.I. and G.W. thank JSPS for Grants-in-Aid for Scientific Research, B (numbers JP17H03104, JP18H01856, JP19H02537) and on Innovative Areas (numbers JP19H05716, JP19H05718).

## Author contributions

T.O. conceived and designed the work, while C.P.Y., N.K., S.K. and T.K. synthesized the compounds. C.P.Y., S.K. and N.K. performed the physicochemical property measurements, single-crystal and thin-film X-ray analyses, and OFET evaluations. H.I. and C.P.Y. calculated the transfer integrals and effective masses. G.W. performed the molecular-dynamics simulations. C.P.Y. performed the DFT studies. C.P.Y., S. K. and T.O. wrote the paper. J.T. and T.O. supervised the work. All authors discussed the results and reviewed the paper prior to submission.

## Competing interests

The authors declare no competing interests.
