## [Peer Review File · Communications Chemistry]

Reviewers' comments:

Reviewer #1 (Remarks to the Author):

The manuscript presented by Okamoto et al. is an interesting piece of work in which the main argument deals with the effect of sterically demanding substituents on the molecular assembly and therefore, on charge transport characteristics of benzoisoquinolinoquinolinetetracarboxylic diimide derivatives.

While the conjugated skeleton studied in this work is not novel (the authors already published it (Sci. Adv. 2020) performing excellent transistor performances), the novelty of this study falls on the use of sterically demanding substituents that allows fine-tuning of the molecular assemblies in the film, which was not possible in their previously published BQQDI derivative due to strong intermolecular interactions in the crystal.

Nowadays, the effect of substituents on already known high-performance conjugated skeletons is of great importance, since subtle modifications can improve and tune device characteristics.

The work presented here is a comprehensive study where it has been demonstrated that an equilibrium between optimal molecular assembly in the single-crystal state and in the thin film is necessary for applications in large-area devices.

There is, however, one aspect of the paper that is not clear for me. The authors compare here OFET performances of single-crystalline thin film devices and polycrystalline devices, observing excellent performances for the polycrystalline films in the case of Cy6-BQQDI.

The lower electrical performance of Ph-BQQDI devices are ascribed to large grain boundary of the polycrystalline thin films. But, how this value compares to electrical performances of polycrystalline films of PhC2-BQQDI (these values are not easy to find either here or in their previous Sci. Adv. paper.)?

Is it not possible the fine tuning of the molecular assemblies in the Ph-BQQDI film as it seems to be in PhC2-BQQDI? A more clear explanation of these aspects should be included in the manuscript.

Otherwise, the discussion seems incomplete.

After these modifications I find this manuscript suitable to be published in CommsChem.

Reviewer #2 (Remarks to the Author):

The manuscript "Controlling molecular assembly and charge transport of n-type organic semiconductors with sterically demanding substituents" examines two BQQDI derivatives with different imide sidechains and studies their molecular assemblies through XRD and, ultimately, relates their solid-state assemblies to their electronic structure and OFET performance. The BQQDI molecules' electronic and physical structures make them good candidates for inclusion as the active layer in OFETs. While the authors' analysis is on BQQDI, its chemical structure is similar to that of PDI, as the authors themselves note. Yet, the authors have not adequately cited relevant works from others that discuss side-chain engineering on PDI derivatives, including those of Frank Wurthner (doi.org/10.1021/acs.chemrev.5b00188) and Ling Zang (doi.org/10.1021/ja061810z). In addition, there are several additional points that need to be addressed before the manuscript can be accepted:

1. Both Cy6-BQQDI and Ph-BQQDI are new molecules. The characterization is insufficient: 1) there are no ¹³C NMR spectra for either compound. The authors should include this characterization, and if they are unable to provide the spectra, then a note should be included as to why; 2) the ¹H NMR

spectra are not clean. For example, in Cy6-BQQDI, there appears to be several peaks indicative of solvent/grease, and the HFIP peaks (between 3.0 – 4.5 ppm) are drowning out the signal from the compound of interest. Can the authors provide a cleaner NMR spectrum? If not, please show a zoomed in region of the aromatic region with integrations as well. Likewise, for the Ph-BQQDI spectrum, please provide a zoomed in region of the aromatic region. The authors should also indicate solvent peaks on the NMR spectra (such as HFIP); and 3) the authors should specify in the Materials and characterization section the ppm shift of HFIP. As of now, it reads “Chemical shifts were reported in parts per million (ppm, d scale) from residual protons in the deuterated solvent for ^1H NMR (5.93 ppm for 1,1,2,2-tetrachloroethane- d_2 (TCE- d_2), 1,1,1,3,3,3-hexafluoro-2-propanol- d_2 (HFIP- d_2) and 7.26 ppm for chloroform- d (CDCl_3)).”

2. The authors base the HOMO and LUMO energy levels entirely from theory. If possible, it would be good to see solid-state frontier orbital measurements; such as, UPS and IPES, to at a minimum have experimental data to support the computations.

3. In lines 58 – 66, the authors discuss the relevance of a deep LUMO energy for ambient stability. Can the authors elaborate more here and state the mechanism? I assume they are referring to a reaction between singlet oxygen and the excited state molecule, but please both elaborate and clarify. It is also unclear what the authors mean about PDI and its relative instability to ambient conditions. Neutral PDI is air stable, thus the authors should clarify this point.

4. In general, the authors refer to hydrogen bonding between a C-H...N and C-H...O. What is the evidence that this is indeed a strong electrostatic interaction? Is the hydrogen atom polarized more in the BQQDI structures than in PDI? Why, then, is it hydrogen bonding?

5. The transfer curves in Figure 5a show significant non-ideality. There are clear “kinks” in the curves and the V_{th} of the OFET consisting of Cy6-BQQDI is high. The cause of these phenomena should be explained. And the authors should follow literature (Nature Materials 2018, 17, 2) to correct for contact resistance in reporting mobility.

6. From the data in Figure 6, it should be straightforward to map the azimuthal distribution of device mobility and quantify its anisotropy to support their finding/conclusion.

7. The contrast between the two derivatives is just two data points. As the authors’ title suggests that sterically demanding side chains can control charge transport, it would be more convincing if there was more than two data points e.g. more than two molecules, in the series. Likewise, there is a causation assigned in the manuscript and title e.g. the side chains control both packing in the solid-state and the OFET mobility. With only two molecules studied, I would avoid such strong causation language.

Reviewer #3 (Remarks to the Author):

In their manuscript “Controlling molecular assembly and charge transport of n-type organic semiconductors with sterically demanding substituents,” T. Okamoto and coworkers report on the synthesis of BQQDI derivatives (Ph-BQQDI and Cy6-BQQDI), the charge-transport capabilities with molecular simulations, and the FET characteristics using the thin films and the single crystals of these organic materials. The FET properties of BQQDI derivatives showed n-type transport, and Cy6-BQQDI exhibited a high mobility in both single-crystalline and polycrystalline thin film FETs. The results demonstrated effective molecular design for molecular assembly, charge transport, suppressing molecular fluctuation, and controlling the molecular assembly.

The reviewer thinks that the conclusions are supported by the analysis of simulations, calculations, and experimental results. However, following issues should be addressed and revised before

consideration of publication.

1. In introduction part, the authors mention that the BQQDI framework has deep-lying LUMO level of -4.17 eV for potential air-stable n-type charge transport, whereas the PDI possesses a shallower LUMO level of -3.80 eV, which is un-stable in air. How difference is the air stability experimentally for the molecules with the LUMO levels between -4.17 eV and -3.80 eV?

2. The mobility was evaluated for single-crystalline transistors using Ph- and Cy6-BQQDI. The higher mobility was recorded in the FET using Cy6-BQQDI than that using Ph-BQQDI, but the V_{th} was comparatively large for the FET using Cy6-BQQDI. Why is the V_{th} higher in Cy6-BQQDI than in Ph-BQQDI? What is the origin of a high V_{th} for Cy6-BQQDI?

3. The authors claim that the mobility of polycrystalline devices of Cy6-BQQDI was improved when a surface modification of substrates was changed from by DTS to by HMDS, because the ratio of face-on/edge-on assemblies decreased for the film on HMDS. Why did the ratio of face-on/edge-on assemblies decrease for the film on HMDS-modified substrate?

4. Why was a higher mobility observed in thin film FET of Cy6-BQQDI with 20 nm-thick-film compared with 40 nm-thick-film?

5. In summary, the authors claim that 'The results... as well as controlling the molecular assembly in the thin-film device state...'. Is the control of the ratio of face-on/edge-on assemblies possible?

6. AFM images of the thin films were shown in Supplementary information (Figures S9 and S10), but these were not referred and commented at all in the text.

7. The numbering of figures in Supplementary Information is wrong; Fig. S14 appears twice.

Point-by-Point Response to Referees

Reviewer #1:

The manuscript presented by Okamoto et al. is an interesting piece of work in which the main argument deals with the effect of sterically demanding substituents on the molecular assembly and therefore, on charge transport characteristics of benzoisoquinolinoquinolinetetracarboxylic diimide derivatives. While the conjugated skeleton studied in this work is not novel (the authors already published it (Sci. Adv. 2020) performing excellent transistor performances), the novelty of this study falls on the use of sterically demanding substituents that allows fine-tuning of the molecular assemblies in the film, which was not possible in their previously published BQQDI derivative due to strong intermolecular interactions in the crystal. Nowadays, the effect of substituents on already known high-performance conjugated skeletons is of great importance, since subtle modifications can improve and tune device characteristics. The work presented here is a comprehensive study where it has been demonstrated that an equilibrium between optimal molecular assembly in the single-crystal state and in the thin film is necessary for applications in large-area devices. There is, however, one aspect of the paper that is not clear for me. The authors compare here OFET performances of single-crystalline thin film devices and polycrystalline devices, observing excellent performances for the polycrystalline films in the case of Cy₆-BQQDI. The lower electrical performance of Ph-BQQDI devices are ascribed to large grain boundary of the polycrystalline thin films. But, how this value compare to electrical performances of polycrystalline films of PhC₂-BQQDI (these values are not easy to find either here or in their previous Sci. Adv. paper.)? Is it not possible the fine tuning of the molecular assemblies in the Ph-BQQDI film as it seems to be in PhC₂-BQQDI? A more clear explanation of these aspects should be included in the manuscript. Otherwise, the discussion seems incomplete. After these modifications I find this manuscript suitable to be published in CommsChem.

Response: We appreciate the thorough analysis by the reviewer. The reviewer also raised an excellent point regarding the comparison between the polycrystalline device performance of PhC₂-BQQDI and Cy₆-BQQDI. To show a clearer comparison between the current Cy₆- and Ph-BQQDI derivatives and the previously reported polycrystalline device of PhC₂-BQQDI, we have added its maximum electron mobility in line 28 of page 9: (eg. μ_{\max} of polycrystalline PhC₂-BQQDI = 0.65 cm² V⁻¹ s⁻¹)³⁸. In reference 38, our group has reported in-depth studies of the polycrystalline device performances of phenylalkyl-substituted BQQDI (including PhC₂-BQQDI) that are fabricated under similar conditions to achieve the optimum

performances. In line 5–7 of page 11, we added a discussion that attributes the high polycrystalline device performance of Cy₆–BQQDI compared with phenylalkyl-substituted BQQDI to its isotropic-like charge-transport capability, where phenylalkyl-substituted BQQDI derivatives show anisotropic charge transports.

The added sentence:

The high polycrystalline device performance of Cy₆–BQQDI despite having the orientationally disordered thin-film assembly could be attributed to its isotropic-like charge-transport capability, which could also explain its superior polycrystalline device performances to those of our previously reported phenylalkyl-substituted BQQDI derivatives fabricated under similar conditions that exhibit more pronounced anisotropic charge-transport behaviors³⁸.

As for whether it is possible to tune the polycrystalline device performance of Ph–BQQDI, we have also deposited the material on DDTS- and HMDS-coated substrates, and no improvement in the device performance was observed. Although we cannot be absolutely certain of the reason that this stage, we suspect that the better polycrystalline device performance of Cy₆–BQQDI is attributed to its charge-transport capability and consistent molecular assembly as its single-crystal structure, compared to the Ph–BQQDI counterpart, which exhibits inconsistent crystal structures in the bulk and polycrystalline forms, and less suitable charge-transport capability. To demonstrate the findings in the manuscript, we have added the following sentence on page 10 lines 7–13. On the other hand, our attempts to improve the polycrystalline device performance of Ph–BQQDI was not successful, as Ph–BQQDI formed the same molecular assembly on dodecyltrimethoxysilane (DDTS) and HDMS-coated substrates as it did on DTS which is inconsistent with its bulk single-crystalline structure (Supplementary Fig. 15b), and no apparent enhancement of the thin-film quality or μ_e was observed for Ph–BQQDI by varying the SAM substrates (Supplementary Fig. 13 and 18).

Supplementary Fig. 18 Transfer characteristics and μ_{sat} of 40 nm-thick Ph-BQQDI polycrystalline OFET on **a** DTS **b** DDTS, and **c** HDMS.

Supplementary Fig. 13 AFM images of 40 nm Ph-BQQDI on **a** DDTS and **b** HDMS.

Supplementary Fig. 15 Out-of-plane polycrystalline thin-film X-ray diffractions of **a** Cy₆-BQQDI on DTS and HMDS, and **b** Ph-BQQDI on DDTS and HDMS.

Reviewer #2:

The manuscript “Controlling molecular assembly and charge transport of n-type organic semiconductors with sterically demanding substituents” examines two BQQDI derivatives with different imide sidechains and studies their molecular assemblies through XRD and, ultimately, relates their solid-state assemblies to their electronic structure and OFET performance. The BQQDI molecules’ electronic and physical structures make them good candidates for inclusion as the active layer in OFETs. While the authors’ analysis is on BQQDI, its chemical structure is similar to that of PDI, as the authors themselves note. Yet, the authors have not adequately cited relevant works from others that discuss side-chain engineering on PDI derivatives, including those of Frank Wurthner (doi.org/10.1021/acs.chemrev.5b00188) and Ling Zang (doi.org/10.1021/ja061810z). In addition, there are several additional points that need to be addressed before the manuscript can be accepted:

Response: We appreciate the reviewer’s positive comments and important suggestions. We thank the reviewer for pointing out important references on PDI that we have missed. The two suggested papers by Frank Wurthner and Ling Zang are now added as references 34 and 31, respectively, on page 2, line 11–12. In addition, PDI-related studies by Zhenan Bao and Jean-Luc Bredas are cited as references 32 and 33, respectively.

1. Both Cy₆-BQQDI and Ph-BQDDI are new molecules. The characterization is insufficient: 1) there are no ¹³C NMR spectra for either compound. The authors should include this characterization, and if they are unable to provide the spectra, then a note should be included as to why; 2) the ¹H NMR spectra are not clean. For example, in Cy₆-BQQDI, there appears to be several peaks indicative of solvent/grease, and the HFIP peaks (between 3.0 – 4.5 ppm) are drowning out the signal from the compound of interest. Can the authors provide a cleaner NMR spectrum? If not, please show a zoomed in region of the aromatic region with integrations as well. Likewise, for the Ph-BQDDI spectrum, please provide a zoomed in region of the aromatic region. The authors should also indicate solvent peaks on the NMR spectra (such as HFIP); and 3) the authors should specify in the Materials and characterization section the ppm shift of HFIP. As of now, it reads “Chemical shifts were reported in parts per million (ppm, d scale) from residual protons in the deuterated solvent for ¹H NMR (5.93 ppm for 1,1,2,2-tetrachloroethane-d₂ (TCE-d₂), 1,1,1,3,3,3-hexafluoro-2-propanol-d₂ (HFIP-d₂) and 7.26 ppm for chloroform-d (CDCl₃)).”

Response: We appreciate the careful examination of our characterization data by the reviewer. The solubility of both Ph- and Cy₆-BQQDI are insufficient to obtain ¹³C NMR, thus, as the reviewer suggested, we added the statement in the compound characterization paragraphs. Owing to the low solubility of Ph- and Cy₆-BQQDI, intensity of the NMR peaks had to be excessively increased and unfortunately, impurity peaks could not be avoided. As suggested by the reviewer, we made the aromatic peaks more visible, and provided zoomed-in images of those peaks. NMR solvents peaks are now identified in the spectra, and the chemical shifts of 1,1,1,3,3,3-hexafluoro-2-propanol-*d*₂ (HFIP-*d*₂) is now provided in ppm.

Modified sections:

A flame-dried Schlenk tube was charged with BQQ-TCDA (117 mg, 0.297 mmol, 1.0 equiv.), aniline (275 mg, 2.96 mmol, 10.0 equiv.), propionic acid (2.00 mL, 29.7 mmol, 100 equiv.), and *o*-DCB (10.0 mL), and the mixture was stirred at 150 °C for 20 h under an atmosphere of argon. The resulting mixture was cooled to room temperature and precipitated in MeOH. The product was collected via vacuum filtration as a red solid (150 mg, 91%). ¹H NMR (400 MHz, TCE-*d*₂): δ 9.74 (s, 2H), 9.39 (d, *J* = 7.2 Hz, 2H), 8.93 (d, *J* = 7.8 Hz, 2H), 7.62-7.53 (m, 6H), 7.38 (d, *J* = 6.8 Hz, 4H). ¹³C NMR spectrum could not be obtained due to insufficient solubility. HRMS (APCI⁺-TOF): Calcd for C₃₄H₂₈N₄O₄ [M+H] 545.1250, found 545.1271. **Elemental Analysis.** Calcd for C₃₄H₂₈N₄O₄: C, 75.00; H, 2.96; N, 10.29. Found: C, 74.92; H, 2.94; N, 10.12.

A flame-dried Schlenk tube was charged with BQQ-TCP (150 mg, 0.184 mmol, 1.0 equiv.), cyclohexylamine (54.6 mg, 0.551 mmol, 3.0 equiv.), and *o*-DCB (6.1 mL) and the reaction mixture was stirred at 150 °C for 1 h under an atmosphere of argon. After that, propionic acid (1.38 mL, 18.4 mmol, 100 equiv.) was added and the mixture was stirred at 150 °C for 3 h. The

resulting mixture was cooled to room temperature and poured into a stirring solution of MeOH. The precipitates were collected via vacuum filtration to give the product as a red-purple solid (96.6 mg, 94%). ¹H NMR (400 MHz, CDCl₃/HFIP-*d*₂): δ 9.58 (s, 2H), 9.28 (d, *J* = 7.8 Hz, 2H), 8.83 (d, *J* = 7.8 Hz, 2H), 5.05-4.96 (m, 2H), 2.53-2.43 (m, 4H), 2.05-1.94 (m, 4H), 1.80-1.77 (m, 6H), 1.64-1.42 (m, 6H). ¹³C NMR spectrum could not be obtained due to insufficient solubility. HRMS (APCI⁺-TOF): Calcd for C₃₄H₂₈N₄O₄ [M+H] 557.2189, found 557.2203. **Elemental Analysis.** Calcd for C₃₄H₂₈N₄O₄: C, 73.37; H, 5.07; N, 10.07, found: C, 73.24; H, 5.13; N, 9.99.

Materials and general characterizations. All amine reagents used in this study were purchased from Tokyo Chemical Industry Co., Ltd and propionic acid was purchased from FUJIFILM Wako Pure Chemical Industries, Ltd without further purifications. *o*-dichlorobenzene (*o*-DCB) was purchased from KANTO chemical Co., Ltd. and purified by a solvent purification system. Starting materials 3,4,9,10-benzo[*de*]isoquinolino[1,8-*gh*]quinolinetetracarboxylic dianhydride (BQQ–TCDA) and 3,9-dimethyl-4,10-bis(2,4,6-trichlorophenyl)benzo[*de*]isoquinolino[1,8-*gh*]quinoline-3,4,9,10-tetracarboxylate (BQQ–TCP) were synthesized and purified in our laboratory prior to this study. All reactions were carried out under an atmosphere of argon. ¹H NMR spectra were recorded on JEOL ECS400 spectrometer (400 MHz). Chemical shifts were reported in parts per million (ppm, δ scale) from residual protons in the deuterated solvent for ¹H NMR (5.93 ppm for 1,1,2,2-tetrachloroethane-*d*₂ (TCE-*d*₂), 3.36 ppm/4.37 ppm for 1,1,1,3,3,3-hexafluoro-2-propanol-*d*₂ (HFIP-*d*₂) and 7.26 ppm for chloroform-*d* (CDCl₃)). The data were presented in the following format: chemical shift, multiplicity (s = singlet, d = doublet, t = triplet, quint = quintet, m = multiplet), coupling constant in Hertz (Hz), signal area integration in natural numbers. Time-of-flight high-resolution mass (TOF-MS) spectrometry measurements were measured on a BRUKER compact-TKP2 mass spectrometer with the atmospheric pressure chemical ionization (APCI) method. Elemental analysis measurements were carried out on a JScience Lab JM10 CHN analyzer at the Comprehensive Analysis Center, the Institute of Scientific and Industrial Research, Osaka University.

Supplementary Fig. 1 The ^1H NMR spectrum of Ph-BQQDI in TCE- d_2 at 100 °C (top), blank TCE- d_2 spectrum at 100 °C (bottom).

Supplementary Fig. 2 The ^1H NMR spectrum of $\text{Cy}_6\text{-BQQDI}$ in $\text{CDCl}_3/\text{HFIP-}d_2$ at room temperature (top), blank $\text{CDCl}_3/\text{HFIP-}d_2$ spectrum at room temperature (bottom).

2. The authors base the HOMO and LUMO energy levels entirely from theory. If possible, it would good to see solid-state frontier orbital measurements; such as, UPS and IPES, to at a minimum have experimental data to support the computations.

Response: We thank the reviewer's excellent suggestion, as we also agree that solid-state LUMO level should be measured for current BQQDI compounds. Unfortunately, due to the lack of UPS or IPES instruments in our group, we could not directly measure the solid-state LUMO levels. We attempted to indirectly obtain the LUMO levels of Ph⁻ and Cy₆-BQQDI by measuring their HOMO band levels by photoelectron yield spectroscopy (PYS) on thin films and calculate the LUMO band levels using their thin-film optical bandgaps. However, we regret to report that inaccurate results were obtained due to relatively deep HOMO levels of BQQDI. Thus, to provide some insights into the electronic structures of Ph⁻ and Cy₆-BQQDI, we have measured their LUMO levels by differential pulse voltammetry (DPV) with ferrocene as a reference in solution, and their data are consistent with the previously reported PhC₂-BQQDI. The following figure is now added in the Supplementary Information, and the experimental procedure for DPV is now added.

Supplementary Fig. 5 Differential pulse voltammetry and their corresponding LUMO energy levels of **a** Ph⁻ and Cy₆-BQQDI, and **b** PhC₂-BQQDI as a reference.

Differential pulse voltammetry (DPV) measurements were performed on an ALS622D Electrochemical Analyzer using glassy carbon as the working electrode, platinum as the counter electrode, and 0.01 M AgNO₃ + 0.1 M tetrabutylammonium hexafluorophosphate (TBAPF₆) in benzonitrile as the reference electrode. Compounds were dissolved in benzonitrile

at 100 °C (< 0.2 mM, saturated solution) and the DPV measurements were performed at the same temperature at a scan rate of 100 mV s⁻¹, with 0.1 M TBAPF₆ as the electrolyte and ferrocene as an internal standard.

3. In lines 58–66, the authors discuss the relevance of a deep LUMO energy for ambient stability. Can the authors elaborate more here and state the mechanism? I assume they are referring to a reaction between singlet oxygen and the excited state molecule, but please both elaborate and clarify. It is also unclear what the authors mean about PDI and its relative instability to ambient conditions. Neutral PDI is air stable, thus the authors should clarify this point.

Response: We thank the reviewer's suggestion to clarify our point on the relationship between LUMO energy and n-type charge transport stability. We have modified the introduction on page 1 lines 29-33 as follows.

One of the challenges associated with the molecular design of n-type OSCs is that **the excited molecules transporting injected charge carriers can be oxidized by ambient singlet oxygen and moisture, which leads to degraded electronic performances in air, thus, the lowest unoccupied molecular orbital (LUMO) level of n-type OSCs should be below -4.0 eV to avoid oxidation of charge carriers and ensure air-stable electron-transporting performances**^{17,18}.

The reviewer is correct that the neutral PDI molecule is chemically stable. We have modified Figure 1 to specify that our discussion is on the stability of charge carriers of PDI during electron transports.

Fig. 1 Molecular features of PhC₂-BQQDI. **a** Structural, packing motif and charge transport comparisons between PDI and BQQDI; **b** Intermolecular interactions of PhC₂-BQQDI; **c** Molecular misalignment of PhC₂-BQQDI.

4. In general, the authors refer to hydrogen bonding between a C-H...N and C-H...O. What is the evidence that this is indeed a strong electrostatic interaction? Is the hydrogen atom polarized more in the BQQDI structures than in PDI? Why, then, is it hydrogen bonding?

Response: We thank the reviewer for this question. Considering the absolute force constants of 2.20 and 7.15 kcal/mol between the BQQDI dimers, the interactions are likely not purely electrostatic driven. According to Thomas Steiner's review on hydrogen bonding in the solid state (*Angew. Chem. Int. Ed.* 2002, 41, 48-76), the force constants and distances between H...N/H...O exhibited between BQQDI dimers fit into the category of weak hydrogen bonding that is contributed by both electrostatic and dispersion forces. As for a comparison of BQQDI and PDI, we have calculated their electrostatic potentials at the B3LYP/6-31+G(d) level of theory, and the results suggested that the hydrogen atoms on the BQQDI π -core are indeed

more polarized than those on PDI. To clarify our statement regarding hydrogen-bonding, Thomas Steiner's review has been added as reference 39.

5. The transfer curves in Figure 5a show significant non-ideality. There are clear “kinks” in the curves and the V_{th} of the OFET consisting of Cy_6 -BQQDI is high. The cause of these phenomena should be explained. And the authors should follow literature (Nature Materials 2018, 17, 2) to correct for contact resistance in reporting mobility.

Response: We thank the reviewer for pointing out this issue. The following sentence is added on page 8 lines 16-1 (page 9). **The large threshold voltage (V_{th}) and the non-ideal transfer curve exhibited by Cy_6 -BQQDI is possible due to the contact resistance attributed the disrupted molecular assembly at the electrode-OSC interface, which leads to a low reliability factor⁴⁴ (r_{sat}) of 0.29 (Supplementary Fig. 10), and an effective μ of $0.67 \text{ cm}^2 \text{ V}^{-1} \text{ s}^{-1}$ (effective $\mu = r_{sat} \times \mu_{claimed}$).**

The following figure has been added to the Supplementary Information showing the extraction of reliability factor for Cy_6 -BQQDI-based OFET.

Supplementary Fig. 10 Reliability factor r_{sat} of Cy₆-BQQDI-based OFET showing the maximum μ_e , where black and gray dashed lines represent the fit to $|I_D|^{1/2}$ and the slope of an electrically ideal OFET, respectively.

6. From the data in Figure 6, it should be straightforward to map the azimuthal distribution of device mobility and quantify its anisotropy to support their finding/conclusion.

Response: We thank the reviewer's comment. Owing to the limited size of the single-crystalline domains of Cy₆-BQQDI, we unfortunately could not measure the azimuthal mobility from *one* single-crystalline domain. Instead, we had to extract the angle-dependent mobility using different single-crystalline domains. We then measured the mobility of single-crystalline Cy₆-BQQDI at 0°, 45°, -45°, and 90° relative to the crystal growth direction [110], where each data point is extracted from 4–5 devices. The overall isotropy suggested by our theoretical calculations is supported by experimental results, since averaged mobilities are found in the range of 1.5–2.0 cm² V⁻¹ s⁻¹. Owing to the small-sized crystals of Ph-BQQDI growth by PVT, we regret to report that its azimuthal distribution of mobility could not be obtained. The following figures are added in Supplementary Information, since the following data is not from the same single-crystalline domain.

Supplementary Fig. 26 The anisotropy of μ_e of single-crystalline Cy₆-BQQDI measured by constructing the OFET channel (channel length = 50 μm) at 0°, 45°, -45°, and 90° relative to the crystal growth direction [110], where each data point is extracted from 4–5 devices. Error bars for angle were set to be $\pm 10^\circ$, and errors bars for mobility were given by standard error.

Supplementary Fig. 27 Representative transfer characteristics of single-crystalline OFETs of Cy₆-BQQDI with channels set along **a** 0° , **b** 90° , **c** 45° , and **d** -45° from the crystal growth direction [100].

The following sentence is added on page 11 lines 4–7. We have further confirmed the isotropic-like charge-transport capability of Cy₆-BQQDI by evaluating the μ_e of its single-crystalline device at 0° , 45° , -45° , and 90° relative to the crystal growth direction [110], and the averaged μ_e are found in the range of $1.5\text{--}2.0$ $\text{cm}^2\text{V}^{-1}\text{s}^{-1}$ (Supplementary Fig. 26 and 27).

7. The contrast between the two derivatives is just two data points. As the authors' title suggest that sterically demanding side chains can control charge transport, it would be more convincing if there was more than two data points e.g. more than two molecules, in the series. Likewise, there is a causation assigned in the manuscript and title e.g. the side chains control both packing in the solid-state and the OFET mobility. With only two molecules studied, I would avoid such strong causation language.

Response: We thank the reviewer's comment. To avoid suggesting overreaching causations, we have modified the title to “**Molecular assembly and charge transport of n-type organic semiconductors with sterically demanding substituents**”

Reviewer #3 (Remarks to the Author):

In their manuscript “Controlling molecular assembly and charge transport of n-type organic semiconductors with sterically demanding substituents,” T. Okamoto and coworkers report on the synthesis of BQQDI derivatives (Ph-BQQDI and Cy₆-BQQDI), the charge-transport capabilities with molecular simulations, and the FET characteristics using the thin films and the single crystals of these organic materials. The FET properties of BQQDI derivatives showed n-type transport, and Cy₆-BQQDI exhibited a high mobility in both single-crystalline and polycrystalline thin film FETs. The results demonstrated effective molecular design for molecular assembly, charge transport, suppressing molecular fluctuation, and controlling the molecular assembly. The reviewer thinks that the conclusions are supported by the analysis of simulations, calculations, and experimental results. However, following issues should be addressed and revised before consideration of publication.

Response: We truly appreciate the positive comments and important suggestions by the reviewer.

1. In introduction part, the authors mention that the BQQDI framework has deep-lying LUMO level of -4.17 eV for potential air-stable n-type charge transport, whereas the PDI possesses a shallower LUMO level of -3.80 eV, which is un-stable in air. How difference is the air stability experimentally for the molecules with the LUMO levels between -4.17 eV and -3.80 eV?

Response: We thank the reviewer for this question. In 2007, Tobin Marks and coworkers reported the air stability of PDI derivatives with different LUMO levels (*J. Am. Chem. Soc.* **2007**, *129*, 15259-15278). OFET based on C₈-substituted PDI with an experimental LUMO level of -3.9 eV showed a mobility of 0.32 cm² V⁻¹ s⁻¹ in vacuum, and the same device showed a mobility of 2 x 10⁻⁴ cm² V⁻¹ s⁻¹ when measured in air. The threshold voltage of the device also increased more than two-fold when measured in air than it was in vacuum. PDI derivatives with LUMO levels below -4.0 eV reported by Mark *et. al.*, as well as BQQDIs reported in the current manuscript exhibit significantly improved stability in mobility and threshold voltages.

2. The mobility was evaluated for single-crystalline transistors using Ph- and Cy₆-BQQDI. The higher mobility was recorded in the FET using Cy₆-BQQDI than that using Ph-BQQDI, but the V_{th} was comparatively large for the FET using Cy₆-BQQDI. Why is the V_{th} higher in Cy₆-BQQDI than in Ph-BQQDI? What is the origin of a high V_{th} for Cy₆-BQQDI?

Response: We thank the reviewer for this comment. A similar question was also raised by reviewer #2 about the large V_{th} of Cy₆-BQQDI. To explain this phenomenon, we have added

the following sentence on page 8 lines 16-1 (page 9). The large threshold voltage (V_{th}) and the non-ideal transfer curve exhibited by Cy₆-BQQDI is possible due to the contact resistance attributed the disrupted molecular assembly at the electrode-OSC interface, which leads to a low reliability factor⁴⁴ (r_{sat}) of 0.29 (Supplementary Fig. 10), and an effective μ of 0.67 cm² V⁻¹ s⁻¹ (effective $\mu = r_{sat} \times \mu_{claimed}$).

3. The authors claim that the mobility of polycrystalline devices of Cy₆-BQQDI was improved when a surface modification of substrates was changed from by DTS to by HMDS, because the ratio of face-on/edge-on assemblies decreased for the film on HMDS. Why did the ratio of face-on/edge-on assemblies decrease for the film on HMDS-modified substrate?
Response: We thank the reviewer for this comment. We are also interested in the substrate dependence of polycrystalline FET properties observed in this study because DTS is generally expected to lead to molecular assemblies more suitable for vacuum deposited OFETs than HMDS. However, this issue is still under investigation.

At this stage, we would like to compare our results with cyclohexyl-substituted naphthalene diimide (NDI) analogue, which only shows the edge-on type assembly (D. Shukla *et al.*, *Chem. Mater.* **2008**, *20*, 7486-7491; T. Kakinuma *et al.*, *J. Mater. Chem. C* **2013**, *1*, 5395-5401). Since they afford the analogous crystal structures with the same space group of C2/m, the difference in polycrystalline orientations could be attributed to interactions between π -conjugated cores. That is, the larger π -conjugated core of BQQDI and the electronegative nitrogen sites incorporated therein could lead to stronger aggregation behaviors and low mobility on SAM substrates compared with NDI. Then, Cy₆-BQQDI would be assembled regardless of the edge-on and face-on orientations if once crystallization occurs on the substrates by vacuum deposition, while NDI favors the edge-on orientation after rearrangements. We would also note that the first layer of some vacuum deposited planar molecules such as pentacene exhibits the face-on orientation on the substrates (S. Mannsfeld *et al.*, *Adv. Mater.* **2009**, *21*, 2294-2298). Thus, we would also assume that Cy₆-BQQDI could behave as such since cyclohexyl groups are not exactly flexible due to its bulkiness sandwiched by two carbonyl groups.

The morphology of vacuum deposited polycrystalline thin films can be affected not only by molecular properties of organic semiconductors but also by relative surface energies between specific crystal surfaces of organic semiconductors and substrates. Therefore, the current issue on the thin film morphology of Cy₆-BQQDI will be further studied by adopting various substrates and comparing Cy₆-BQQDI analogues with functionalized cyclohexyl groups, which will be reported in the future.

4. Why was a higher mobility observed in thin film FET of Cy₆-BQQDI with 20 nm-thick-film compared with 40 nm-thick-film?

Response: We thank the reviewer for this comment. From the AFM images of Cy₆-BQQDI with 20 nm- and 40 nm-thick films (Supplementary Fig. 12), grain boundaries are visibly larger in 40 nm films than they are in 20 nm-thick films. Thus, the thin-film quality of Cy₆-BQQDI can be improved by controlling the film thickness, which leads to improved polycrystalline device performance. We have added the following sentence on page 9 lines 32–33. We found that by reducing the Cy₆-BQQDI OSC layer thickness from 40 nm to 20 nm on HMDS, an excellent highest μ_e of $1.0 \text{ cm}^2 \text{ V}^{-1} \text{ s}^{-1}$ could be achieved, and the μ_e appeared to be independent of the channel length (100-500 μm) (Supplementary Fig. 20–22), which was likely due to the improved thin-film quality and smaller grain boundaries shown by the AFM images as we reduced the thickness of the polycrystalline films (Supplementary Fig. 12), though the exact origin of this phenomenon requires further investigations.

5. In summary, the authors claim that ‘The results... as well as controlling the molecular assembly in the thin-film device state...’. Is the control of the ratio of face-on/edge-on assemblies possible?

Response: We thank the reviewer for the comment. Our intention with that sentence was to highlight the tuning of thin-film molecular assembly by controlling the substituents of BQQDI. From our current results, we have gathered evidence which suggests that both single-crystalline and polycrystalline assemblies in the thin-film state can be tuned by changing the substituent from phenyl to cyclohexyl on the BQQDI π -core. Although, to avoid confusion, we modified the above-mentioned sentence as follows: The results herein demonstrate an effective molecular design for molecular assembly, charge transport, and suppressing molecular fluctuations in the bulk single-crystal state, as well as controlling the molecular assembly in the thin-film device state for achieving the optimum charge-transport capabilities via substituent engineering.

6. AFM images of the thin films were shown in Supplementary information (Figures S9 and S10), but these were not referred and commented at all in the text.

Response: We thank the reviewer for the important comment. Figure S9 (now Supplementary Fig. 11) is now mentioned on page 9 lines 22–24. “...shown by the atomic force microscope (AFM), whereas Cy₆-BQQDI forms smaller grain boundaries and better quality of polycrystalline thin films (Supplementary Fig. 11).” In view of the second comment, we now

mentioned Supplementary Fig. 12 on page 9 line 32. "...which was likely due to the improved thin-film quality and smaller grain boundaries shown by the AFM images as we reduced the thickness of the polycrystalline films (Supplementary Fig. 12)."

7. The numbering of figures in Supplementary Information is wrong; Fig. S14 appears twice.

Response: We thank the reviewer for the careful examination of our manuscript. The first Fig. S14 is now changed to Supplementary Fig. 17, and the second Fig. S14 is changed to Supplementary Fig. 20.

The manuscript “Controlling molecular assembly and charge transport of n-type organic semiconductors with sterically demanding substituents” examines two BQQDI derivatives with different imide sidechains and studies their molecular assemblies through XRD and, ultimately, relates their solid-state assemblies to their electronic structure and OFET performance. The BQQDI molecules’ electronic and physical structures make them good candidates for inclusion as the active layer in OFETs. The authors have thoroughly addressed the concerns of all the reviewers and, thus, have substantially improved the manuscript. I have a couple additional points that require attention, and if these queries are addressed, then I believe this manuscript should be accepted for publication.

1. Figure 4 is difficult to interpret and what “B” represents is unclear. I see this is defined in the SI, and the authors should move the definition of “B” to the manuscript, or at a minimum, define “B” in the main text/caption. In Figure 4, the scale bars should be labeled to indicate what the colors represent. I believe this is molecular fluctuations, so at a minimum, please define in the figure caption and label the figure and scale bars more clearly.
2. The authors put degrees “°” instead of “Å” on page 9, line 12. It is highlighted below. “The deposited thin film of Ph–BQQDI does not assume its single-crystal structure, as the d-spacing of 19.5 Å at $2\theta = 4.52^\circ$ (d-spacing = 15.5° in single crystal) indicates a tilting angle of 24.5° between the long axis of the molecules and the substrate (Supplementary Fig. 14 and 16), which possibly originates from the interactions between the substrate and OSC molecules.”
3. On the above point, the authors suggest the molecular orientation relative to the substrate is at an angle of 24.5° given the d-spacing, yet this is an oversimplification, as the

molecular orientation could potentially be anything. There has been no fitting or additional evidence to suggest the molecules simply tilted (relative to their single crystal orientation) to yield this d-spacing, so I believe it is just a hypothesis. Thus, at a minimum, please state this is a hypothesis. Additionally then, the complementary SI Figure is misleading, as it suggests this tilted orientation has been proven, so please again state this is a hypothesis or one interpretation in the SI.

4. The authors discussion of grain boundaries is unclear. First, I do not understand if “large grain boundaries” means a high density of grain boundaries or they simply mean large grains. Please amend the language appropriately. Second, the AFM images are not convincing evidence to support grain boundary density. Figure SI 11a and 11b are on different lengths scales, so a comparison is difficult. Figure S11a is also blurry, thus it is not obvious what is terracing or what are potentially grain boundaries. Did the authors perform optical microscopy? If so, this could provide evidence of grain boundary density that could help make their arguments. The text is highlighted below.

“We evaluated the polycrystalline thin-film OFETs of Ph- and Cy6-BQQDI, and the highest μ_e of $0.16 \text{ cm}^2 \text{ V}^{-1} \text{ s}^{-1}$ was obtained for Ph-BQQDI (Supplementary Fig. 18a), which is one-order lower than its single-crystalline device, likely due to large grain boundaries of the polycrystalline thin films shown by the atomic force microscope (AFM), whereas Cy6-BQQDI forms smaller grain boundaries and better quality of polycrystalline thin films (Supplementary Fig. 11).”

Reviewers' comments:

Reviewer #1 (Remarks to the Author):

The authors have revised the manuscript according to my previous comments, so I think it is now suitable for publication

Reviewer #2 (Remarks to the Author):

[Editorial Note: Please see attached report.]

Reviewer #3 (Remarks to the Author):

I appreciate the authors' response. The authors responded to my previous concerns appropriately in the revised manuscript. I would like to recommend the manuscript for publication in Communications Chemistry.

Reviewers' comments:

Reviewer #2 (Remarks to the Author):

The manuscript "Controlling molecular assembly and charge transport of n-type organic semiconductors with sterically demanding substituents" examines two BQQDI derivatives with different imide sidechains and studies their molecular assemblies through XRD and, ultimately, relates their solid-state assemblies to their electronic structure and OFET performance. The BQQDI molecules' electronic and physical structures make them good candidates for inclusion as the active layer in OFETs. The authors have thoroughly addressed the concerns of all the reviewers and, thus, have substantially improved the manuscript. I have a couple additional points that require attention, and if these queries are addressed, then I believe this manuscript should be accepted for publication.

1. Figure 4 is difficult to interpret and what "B" represents is unclear. I see this is defined in the SI, and the authors should move the definition of "B" to the manuscript, or at a minimum, define "B" in the main text/caption. In Figure 4, the scale bars should be labeled to indicate what the colors represent. I believe this is molecular fluctuations, so at a minimum, please define in the figure caption and label the figure and scale bars more clearly.

2. The authors put degrees "°" instead of "Å" on page 9, line 12. It is highlighted below. "The deposited thin film of Ph-BQQDI does not assume its single-crystal structure, as the d-spacing of 19.5 Å at $2\theta = 4.52^\circ$ (d-spacing = 15.5° in single crystal) indicates a tilting angle of 24.5° between the long axis of the molecules and the substrate (Supplementary Fig. 14 and 16), which possibly originates from the interactions between the substrate and OSC molecules."

3. On the above point, the authors suggest the molecular orientation relative to the substrate is at an angle of 24.5° given the d-spacing, yet this is an oversimplification, as the molecular orientation could potentially be anything. There has been no fitting or additional evidence to suggest the molecules simply tilted (relative to their single crystal orientation) to yield this d-spacing, so I believe it is just a hypothesis. Thus, at a minimum, please state this is a hypothesis. Additionally then, the complementary SI Figure is misleading, as it suggests this tilted orientation has been proven, so please again state this is a hypothesis or one interpretation in the SI.

4. The authors discussion of grain boundaries is unclear. First, I do not understand if "large grain boundaries" means a high density of grain boundaries or they simply mean large grains. Please amend the language appropriately. Second, the AFM images are not convincing evidence to support grain boundary density. Figure SI 11a and 11b are on different lengths scales, so a comparison is difficult. Figure S11a is also blurry, thus it is not obvious what is terracing or what are potentially grain boundaries. Did the authors perform optical microscopy? If so, this could provide evidence of grain boundary density that could help make their arguments. The text is highlighted below.

"We evaluated the polycrystalline thin-film OFETs of Ph- and Cy6-BQQDI, and the highest μ_e of 0.16 cm² V⁻¹ s⁻¹ was obtained for Ph-BQQDI (Supplementary Fig. 18a), which is one-order lower than its single-crystalline device, likely due to large grain boundaries of the polycrystalline thin films shown by the atomic force microscope (AFM), whereas Cy6-BQQDI forms smaller grain boundaries and better quality of polycrystalline thin films (Supplementary Fig. 11)."

Point-by-Point Response to Referees

Reviewer #2:

The manuscript “Controlling molecular assembly and charge transport of n-type organic semiconductors with sterically demanding substituents” examines two BQQDI derivatives with different imide sidechains and studies their molecular assemblies through XRD and, ultimately, relates their solid-state assemblies to their electronic structure and OFET performance. The BQQDI molecules’ electronic and physical structures make them good candidates for inclusion as the active layer in OFETs. The authors have thoroughly addressed the concerns of all the reviewers and, thus, have substantially improved the manuscript. I have a couple additional points that require attention, and if these queries are addressed, then I believe this manuscript should be accepted for publication.

Response: We again appreciate the positive comments and thorough examination by the reviewer. We have revised the manuscript as suggested. During the revision process, we have found that the Cy₆-BQQDI single-crystal structure exhibits static disordering which can lead to different assignments of nitrogen atoms in the *bay* positions. Although they can be randomly arranged in the actual structure, two types of periodic structures, namely, the A- and B-forms are considered for our calculations. We calculated their transfer integrals and effective masses, both forms show similar values. What is different is their molecular fluctuations from MD simulations, where the A-form exhibits small B-factors where the B-form exhibits larger B-factors. Owing to these interesting findings, we calculated their variant transfer integral values based on MD simulations to see the effect of molecular fluctuations on their charge-transport capabilities. Surprisingly, both A- and B-forms exhibited similar variant transfer integrals despite showing different B-factors, whereas Ph-BQQDI shows more pronounced changes in its transfer integrals as a result of molecular fluctuations. The current results may suggest that Cy₆-BQQDI with the isotropic charge transport is rather insensitive towards molecular fluctuations compared to the anisotropic Ph-BQQDI, which is in agreement with the report by Troisi *et al.* (*Nature Materials*, 2017). The major changes made in this manuscript are listed in the last page.

1. Figure 4 is difficult to interpret and what “B” represents is unclear. I see this is defined in the SI, and the authors should move the definition of “B” to the manuscript, or at a minimum, define “B” in the main text/caption. In Figure 4, the scale bars should be labeled to indicate

what the colors represent. I believe this is molecular fluctuations, so at a minimum, please define in the figure caption and label the figure and scale bars more clearly.

Response: We are thankful for the reviewer's comment and suggestion. As the reviewer commented, the B -factor represents thermal fluctuations of each atom in the aggregated structure, which is conventionally used in crystallography of protein. We have now labelled the scale bar in Figure 4 and added the unit of B -factor in the captions.

Fig. 4 Molecular dynamics simulations of Ph-BQQDI and Cy₆-BQQDI. **a** *Ortho/bay* positions of BQQDI and color-coded B -factor ($\text{\AA}^2 \text{s}^{-1}$) distributions obtained from the trajectories during the last 10 ns of a 100 ns MD simulations in the NTP ensemble (the magnitude of B -factors is represented by the color-coded scale bar ranging from blue (small value) to red (large value)). **b** Variant t_1 and t_2 distributions and standard deviations (σ) calculated from 100 pairs of adjacent dimers revealing the magnitude of the molecular fluctuations.

We have added a brief definition of B -factors on page 7 lines 13–15 and moved the molecular dynamics calculation method and mathematical definition of B -factors in the Method section as follows. Discussion on variant transfer integral distribution is also added.

Both Ph-BQQDI and Cy₆-BQQDI show small B -factors, which is the temperature dependence of thermal atomic fluctuations (see Method section for the mathematical definition)

We picked up 100-200 pairs of adjacent dimers in the π - π stacking directions from the MD-simulated Ph- and Cy₆-BQQDI and calculated variant t_1 and t_2 distributions and standard deviations (σ) to reveal the effect of molecular fluctuations on their charge-transport capabilities. Ph-BQQDI has an averaged $t_1 = +59.7$ meV and $t_2 = +34.2$ meV, with corresponding σ of 24.2 and 11.8 meV, respectively (Fig. 4b). To our surprise, despite having completely different B -factors, A- and B-forms of Cy₆-BQQDI demonstrate very similar variant t values in the π - π stacking directions. The A-form is showing averaged $t_1 = +66.7$ meV and $t_2 = +69.1$ meV, with σ of 16.6 and 16.9 meV (Fig. 4b). The B-form shows averaged $t_1 = +68.1$ meV and $t_2 = +69.5$ meV that are similar to those of the A-form despite the former's large B -factors. The σ of averaged t_1 and t_2 of the B-form are calculated to be 21.6 and 24.1 meV (Supplementary Fig. 9). It has been reported that the ratio of σ and averaged t values (σ/t_{Avg}) quantifies the effect of molecular fluctuations on charge transport⁴³. Ph-BQQDI exhibits σ/t_{Avg} of 0.41 and 0.35 in t_1 and t_2 directions, respectively, and Cy₆-BQQDI demonstrates smaller σ/t_{Avg} of 0.25 and 0.24 for the A-form, and 0.32 and 0.35 for the B-form, in t_1 and t_2 directions, respectively. The current calculations suggest that the charge-transport capability of Ph-BQQDI is strongly affected by molecular fluctuations compared to Cy₆-BQQDI. Troisi *et. al* reported that large isotropic t values in the 2D herringbone assembly can be insensitive towards dynamic fluctuations⁴¹. Our results here may suggest that the isotropic t values of Cy₆-BQQDI in the 2D brickwork assembly are also effective against dynamic fluctuations.

Molecular dynamics simulations. Molecular dynamics (MD) simulations of single crystal structures in this study were carried out by using the MD program GROMACS 2016.3. The number of molecules, temperature, and the size and the shape of the initial MD cell of Ph-BQQDI and Cy₆-BQQDI are listed in the Supporting Information. Since the intra- and interatomic interactions should be treated explicitly for analyzing the atomistic dynamics, an all-atom model was employed in accordance with generalized Amber force field parameters⁵¹.

The partial atomic charges of the simulated molecules were calculated using the restrained electrostatic potential (RESP)⁵² methodology, based on DFT calculations with the 6-31G(d) basis set using the GAUSSIAN 09 revision E01 program⁵⁰.

For each system, the pre-equilibration run was initially performed at the given temperature for 5 ns after the steepest descent energy minimization. All systems were subjected to pre-equilibration runs in the NTV (constant number of substances (N), constant temperature (T), constant volume (V)) ensemble before their equilibration runs. During the pre-equilibration runs for the NTV ensemble, the Berendsen thermostat⁵³ was used to maintain the temperature of the system with relaxation time of 0.2 ps and the volume of the MD cell was kept constant. Subsequently, the NTP ensemble the equilibration run was performed using the Nosé-Hoover thermostat⁵⁴⁻⁵⁶ and Parrinello-Rahman barostat⁵⁷ with relaxation times of 1.0 and 5.0 ps, respectively. For all MD simulations in the NTP ensemble, the pressure of the system was kept at 1.0 bar. The smooth particle-mesh Ewald (PME)⁵⁸ method was employed to treat the long-range electrostatic interactions and the real space cutoff and the grid spacing are 1.2 and 0.30 nm, respectively. The time step was set to 1 fs.

To compare temperature dependence of thermal atomic fluctuations between different molecules, we calculated the B-factors related to the thermal stability as expressed below:

$$B = \frac{8}{3} \pi^2 \Delta_i^2$$

where Δ_i is the root mean square fluctuations (RMSF) of atom i . The RMSF values can be estimated by using following equation:

$$\Delta_i = \sqrt{\frac{1}{T} \sum_{j=1}^T |\mathbf{r}_i(t_j) - \bar{\mathbf{r}}_i|^2}$$

where T is the number of steps, $\mathbf{r}_i(t_j)$ is the position coordinate of atom i , and $\bar{\mathbf{r}}_i$ is the average of $\mathbf{r}_i(t_j)$ during T . The RMSF values were analyzed from MD trajectories during the last 10 ns in the equilibrium.

More than one hundred pairs of dimers from the MD simulated molecules are picked up to calculate their t values in π - π stacking directions in response to molecular fluctuations.

Variant t_1 and t_2 values as well as their σ are calculated to show the effect of molecular fluctuations on charge-transport capabilities

2. The authors put degrees “°” instead of “Å” on page 9, line 12. It is highlighted below. “The deposited thin film of Ph-BQQDI does not assume its single-crystal structure, as the d-spacing of 19.5 Å at $2\theta = 4.52^\circ$ (d-spacing = 15.5° in single crystal) indicates a tilting angle of 24.5° between the long axis of the molecules and the substrate (Supplementary Fig. 14 and 16), which possibly originates from the interactions between the substrate and OSC molecules.”

Response: We again thank the reviewer for the careful examination of our manuscript. The mistake is now corrected on page 10 lines 12–14 as follows.

The deposited thin film of Ph-BQQDI does not assume its single-crystal structure, as the polycrystalline d -spacing of 19.5 Å at $2\theta = 4.52^\circ$ differs from its single-crystalline d -spacing of 15.5 Å.

3. On the above point, the authors suggest the molecular orientation relative to the substrate is at an angle of 24.5° given the d-spacing, yet this is an oversimplification, as the molecular orientation could potentially be anything. There has been no fitting or additional evidence to suggest the molecules simply tilted (relative to their single crystal orientation) to yield this d-spacing, so I believe it is just a hypothesis. Thus, at a minimum, please state this is a hypothesis. Additionally then, the complementary SI Figure is misleading, as it suggests this tilted orientation has been proven, so please again state this is a hypothesis or one interpretation in the SI.

Response: We thank the reviewer for this comment. We now specify in both the main text on page 10 lines 14–16 and SI that the tilting angle is our hypothesis.

A hypothesized tilting angle between the long axis of the molecules and the substrate is 24.5° based on the longest intramolecular H···H distance (21.47 Å based on the single-crystal structure) of Ph-BQQDI (Supplementary Fig. 16 and 18), which possibly originates from the interactions between the substrate and OSC molecules.

Supplementary Fig. 18 The longest intramolecular H...H distance (molecular length) in the single crystal structure.

4. The authors discussion of grain boundaries is unclear. First, I do not understand if “large grain boundaries” means a high density of grain boundaries or they simply mean large grains. Please amend the language appropriately. Second, the AFM images are not convincing evidence to support grain boundary density. Figure SI 11a and 11b are on different lengths scales, so a comparison is difficult. Figure S11a is also blurry, thus it is not obvious what is terracing or what are potentially grain boundaries. Did the authors perform optical microscopy? If so, this could provide evidence of grain boundary density that could help make their arguments. The text is highlighted below.

“We evaluated the polycrystalline thin-film OFETs of Ph- and Cy6-BQQDI, and the highest μ_e of 0.16 cm² V⁻¹ s⁻¹ was obtained for Ph-BQQDI (Supplementary Fig. 18a), which is one-order lower than its single-crystalline device, likely due to large grain boundaries of the polycrystalline thin films shown by the atomic force microscope (AFM), whereas Cy6-BQQDI forms smaller grain boundaries and better quality of polycrystalline thin films (Supplementary Fig. 11).”

Response: We thank the reviewer for raising this important point. The grain boundary density is indeed an important parameter for investigating polycrystalline thin films, however, as the reviewer stated, it is difficult to exactly quantify grain boundary density with AFM. After consideration, we think it is more appropriate to mainly attribute the low polycrystalline μ_e of Ph-BQQDI to the different molecular assembly the polycrystalline thin film adopts from its single-crystal structure, and its poorer crystallinity in polycrystalline thin films. As a better comparison, we provided AFM images of Ph- and Cy6-BQQDI polycrystalline thin films with the same length scale. Though we cannot quantify their grain boundary densities, it is

reasonable to estimate their grain sizes from the current AFM images. Thus, we have modified the above sentence as follows.

We evaluated the polycrystalline thin-film OFETs of Ph- and Cy₆-BQQDI, and the highest μ_e of $0.16 \text{ cm}^2 \text{ V}^{-1} \text{ s}^{-1}$ was obtained for Ph-BQQDI (Supplementary Fig. 20a), which is one-order lower than its single-crystalline device. **Although the critical reason has not been clarified, we hypothesize that the inconsistent polycrystalline thin-film assembly with its single-crystal structure possibly leads to a less electron-transport capability than expected from the single crystal structure. In addition, the surface morphology of Ph-BQQDI with less significant terracing structure than that of the Cy₆-BQQDI thin film despite comparable grain sizes (>500 nm) implies lower crystallinity of Ph-BQQDI thin films (Supplementary Fig. 14).**

Supplementary Fig. 14 AFM images of **a** Ph-BQQDI and **b** Cy₆-BQQDI, on DTS with thin-film thickness of 40 nm.

List of Major Changes Made in the Manuscript

Fig. 1c Specific molecular displacements and transfer integrals of PhC₂-BQQDI are added.

Fig. 3 Specific molecular displacements of Ph- and Cy₆-BQQDI are added

Page 9 lines 8–11. Static disordering of the Cy₆-BQQDI single-crystal structure and its B-form are introduced.

Page 9 lines 25–27 Transfer integrals of Cy₆-BQQDI (B-form) are listed. Intermolecular distances, molecular displacements, effective mass are shown in Supplementary Fig. 8.

Fig. 4 *bay/ortho* positions of BQQDI are shown, the unit of B-factors is added. Variant transfer integrals are added in **b**.

Discussion MD simulations of Cy₆-BQQDI (A- and B-forms) as well as variant transfer integrals are added on page 8 lines 6–31.

Variant transfer integral distributions of Cy₆-BQQDI (B-form) are shown in Supplementary Fig. 9.

Page 10 lines 12–14 are corrected according to reviewer 2's comments.

Discussion on the polycrystalline thin films on page 10 lines 24–29 are corrected according to reviewer 2's comments.